# DeepDiver: Adaptive Web-Search Intensity Scaling via Reinforcement Learning

**Wenxuan Shi,[*] Haochen Tan[*†], Chuqiao Kuang, Xiaoguang Li,**
**Hanting Chen, Xiaozhe Ren, Yasheng Wang, Lu Hou, Lifeng Shang**

Huawei Language Model Lab
{wenxuan.shi, haochen.tan}@huawei.com

## Abstract

Information seeking demands iterative evidence gathering and reflective reasoning, yet large language models (LLMs) still struggle with it in open-web question answering. Existing prompting and supervised fine-tuning (SFT) methods remain fixed by prompt rules or training corpora, and are usually benchmarked only on well-structured wiki sources, limiting real-world adaptability. We introduce **WebPuzzle**, a 24k-sample training and 275-sample test benchmark that evaluates information seeking on the live internet, across both wiki and open-domain queries. Leveraging 7k WebPuzzle instances, we develop **DeepDiver**, a reinforcement-learning (RL) framework that cultivates **Search Intensity Scaling (SIS)**—an emergent ability to escalate search frequency and depth instead of settling on overconfident, under-evidenced answers. With SIS, Qwen2.5-7B-Instruct and Pangu-7B-Reasoner attain performance on real-web tasks comparable to the 671B-parameter DeepSeek-R1. We detail DeepDiver's curriculum from cold-start SFT to a well designed RL procedure, and show that its seeking policy generalized from closed-ended queries to open-ended generation such as long-form writing. Our results advance adaptive information seeking in LLMs and provide a rigorous benchmark for future work.

## 1 Introduction

*Information seeking* [34] is a fundamental cognitive skill that involves iterative evidence gathering, reflective reasoning, and the resolution of conflicting information. Despite significant advancements in artificial intelligence, LLMs continue to struggle with replicating such information-seeking behaviors. Knowledge-intensive question answering, a central challenge for LLMs, requires a robust capability for information seeking. Current models often fail to determine when and what information to seek, verify the relevance of evidence, and reason effectively over noisy or conflicting contexts.

Iterative Retrieval-Augmented Generation (RAG) [15] frameworks have been proposed to address these challenges by alternating between retrieval and reasoning. Existing approaches generally fall into two categories: prompting-based and task-specific supervised fine-tuning (SFT). Prompting-based methods leverage predefined rules or in-context learning (ICL) [3], forcing the LLM to follow a specific pipeline to complete complex tasks [11, 23, 32, 42, 16]. However, the fixed reasoning flow provided in the prompts limits their adaptability to complex, dynamic problems. In contrast, supervised fine-tuning methods train models to improve retrieval and reasoning capabilities [2, 41], generally yielding better performance. However, these methods often internalize inference patterns tied to the training corpus, restricting generalization to more dynamic or unseen situations.

Recently, reinforcement learning (RL) [26, 13] has been applied to enhance inference-time reasoning in LLMs, enabling iterative refinement and exploration of reasoning [6, 31, 19]. Several studies

---

[*]Equal contribution.
[†]Corresponding author.

39th Conference on Neural Information Processing Systems (NeurIPS 2025).

have integrated RL into iterative RAG frameworks, encouraging models to explore diverse reasoning paths and rewarding accurate outcomes [12, 25, 4, 45]. However, these works predominantly train and evaluate their methods on well-structured datasets such as HotpotQA [39], which are based on corpora like Wikipedia. In such settings, many tasks can be effectively solved using the LLMs' internal knowledge, and the introduced search environments are "clean," containing minimal noise or conflicting information. In contrast, real-world search environments are inherently more complex—characterized by noisy, inconsistent, and unreliable sources. This discrepancy limits the generalizability of the reported "incentivized search capabilities" to more realistic, open-ended information-seeking scenarios.

To investigate RL-guided LLM behaviors in more realistic, open-domain scenarios, we introduce **WebPuzzle**, a dataset designed to evaluate information-seeking capabilities in real-world search environments. WebPuzzle contains 24k training samples and 275 human-annotated test examples, covering tasks solvable with Wikipedia content as well as broader open-domain queries extracted from open-web environment. Even Wikipedia subset are rigorously validated to require external retrieval, ensuring a realistic assessment of LLMs' search behaviors. Along with WebPuzzle, we introduce **DeepDiver**, an RL-driven search and reasoning framework trained on this dataset. DeepDiver interacts with real-world search engines, continuously refining and denoising retrieved documents to provide accurate answers. A key innovation of DeepDiver is the emergent capability of **search intensity scaling (SIS)**, which dynamically scales up the search frequency and depth as information demands increase. This enables LLMs to tackle more complex, information-intensive problems under open-web environment. Together, WebPuzzle and DeepDiver provide a comprehensive framework for developing and examining information seeking ability of LLMs, offering a promising approach for knowledge-intensive problem solving.

Through systematic empirical analysis, we identify critical factors that influence model behavior, including the search intensity, the training environment, and the generalization capabilities. Our analysis reveals several key insights: (1) DeepDiver exhibits exceptional information-seeking ability via adaptive SIS, where the depth and frequency of searching proportional to both problem difficulty and the model's performance. (2) Compared to the "clean" Wiki-based environment, WebPuzzle and real-world search settings better support complex reasoning beahviours, guiding LLMs to actively supplement evidence, resolve conflicts, verify content, and reflect for self-correction. (3) RL training significantly enhances the generalization capability of LLMs, enabling the transition from closed-ended to open-ended problems. In conclusion, our method underscores the potential of reinforcement learning to foster emergent adaptive search behaviors—specifically, search intensity scaling—in LLMs. This significantly enhances their ability to perform adaptive, verifiable, and scalable information seeking, providing a promising direction for future advancements in knowledge-intensive problem solving.

## 2 Preliminaries

### 2.1 Iterative RAG

We formulate the iterative Retrieval-Augmented Generation (RAG) framework for question answering. Given a question $q$, the model iteratively performs reasoning and retrieval to produce an answer.

At each iteration $t \in \{1, 2, \ldots, T\}$, the model maintains a reasoning history $\mathcal{H}_{t-1} = \{q, (r_1, s_1, d_1), \ldots, (r_{t-1}, s_{t-1}, d_{t-1})\}$, where $r_i$ represents the intermediate CoT generated at round $i$, $s_i$ denotes search queries, and $d_i$ denotes retrieved documents from web search.

At round $t$, conditioned on history $\mathcal{H}_{t-1}$, the model first generates intermediate reasoning $r_t \sim p(r_t \mid \mathcal{H}_{t-1})$ to analyze the current status. Then, based on the reasoning $r_t$, the model selects one of two actions: (1) **Search:** generate additional queries $s_t \sim p(s_t \mid \mathcal{H}_{t-1}, r_t)$ and retrieve supporting documents $d_t = \text{Retrieval}(s_t)$; (2) **Answer:** finalize the answer $a \sim p(a \mid \mathcal{H}_{t-1}, r_t)$ to question $q$. This iterative reasoning-and-retrieval process continues until the model chooses the answer action, resulting in a final answer that is well supported by retrieved evidence and explicit reasoning steps.

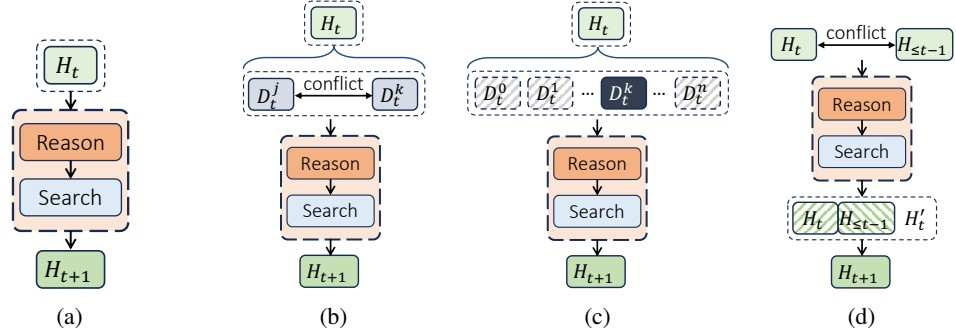

Figure 1: Illustration of four key information-seeking behaviors: (a) *Evidence Gathering & Supplements* (b) *Conflict Resolution* (c) *Verification & Denoising* and (d) *Reflection & Correction*.

## 2.2 Information Seeking Behaviour

We define information seeking behaviour within iterative RAG frameworks as a structured decision-making process: at each iteration the model adopts specific strategies to resolve uncertainties, improve evidence quality, and enhance the overall reliability of answers. Formally, at iteration $t$, conditioned on the reasoning history $\mathcal{H}_{t-1}$ and current intermediate reasoning $r_t$, the model exhibits several strategies to guide its search and reasoning processes.

Inspired by the findings of Gandhi et al. [7], we categorize these strategies into four types of information seeking behaviours: (1) *Evidence Gathering & Supplements*, where the model actively seeks to fill identified knowledge gaps by formulating targeted queries $s_t$ and retrieving supporting documents $d_t$, formally represented as $(s_t, d_t) \sim p(s_t, d_t \mid \mathcal{H}_{t-1}, r_t)$, where $d_t = \text{Retrieval}(s_t)$. This strategy is exemplified by traditional question-answering datasets such as 2Wiki [9], HotpotQA [39], and FRAMES [14]; (2) *Conflict Resolution*, where the model reasons about inconsistencies and evaluates competing claims when retrieved information contains contradictions; (3) *Verification & Denoising*, where the model cross-checks facts and isolates trustworthy information from noisy or irrelevant retrieved content; and (4) *Reflection & Correction*, where the model periodically re-assesses its reasoning trajectory, revisits earlier assumptions, and explicitly corrects previous reasoning steps for iterative refinement. The latter three behaviours can be represented generally as generating reasoning steps $r_t \sim p(r_t \mid \mathcal{H}_{t-1}, d_t)$, where the specific conditions (e.g. presence of contradiction, noise, or previous mistakes) differ according to each behaviour.

Existing works adopting the wiki-based datasets, limiting their scope to structured and well-organized knowledge bases, and thus predominantly emphasize "Evidence Gathering & Supplements". To prove this observation, we show an detailed analysis in Appendix A.2. In contrast, our proposed **WebPuzzle** and the real-world searching environment explicitly necessitates employing all four behaviours, thereby reflecting a more comprehensive and realistic scenario for real-world problem-solving with web-searching. More details about WebPuzzle will be included in section 3.1.

## 3 Method

In this section, we discuss the details of our approach. We begin by introducing WebPuzzle, a dataset designed to address real-world reasoning and search challenges. Next, we describe DeepDiver, a reinforcement learning-based training framework aimed at enhancing LLMs with robust capabilities introduced in section 2.2.

### 3.1 WebPuzzle

Unlike existing open-domain QA datasets based on Wikipedia [39, 21, 14] where LLMs often perform well using only internal knowledge, we introduce **WebPuzzle**, a dataset designed to evaluate LLMs' ability to locate and reason over noisy, scattered information on the open web. Figure 2 illustrates our data synthesis and curation processes.

**Candidate Data Generation** We collect candidate data from Wiki-corpus and real-user queries with retrieved webpages from our deployed smart assistant service. Our generation involves two

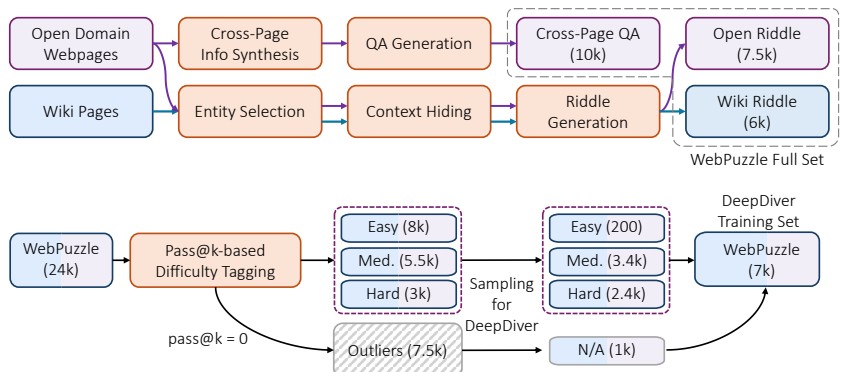

Figure 2: WebPuzzle pipeline. Above: *Candidate Generation:* Wiki and open-web pages yield QA pairs via (i) Cross-Page QA and (ii) Riddle pipelines, grouped as Cross-Page QA, Open Riddle, and Wiki Riddle. Below: *Difficulty Tagging:* Each sample is tagged (easy/medium/hard) for adaptive mixing in RL; DeepDiver is trained on a curated 7k-sample mix.

approaches: (1) Cross-page question generation, where an LLM extracts facts from web pages to generate "inverted" questions, answers and checklists [29, 14]—applied only to open web pages as Wiki-corpus tends to produce overly-simple questions; and (2) Riddle creation, where the LLM selects distinctive entity attributes and applies **obfuscation or generalization** to create challenging problems, with original entities as labels. Examples appear in Appendix D.1. More quality assurance protocols are shown in Appendix D.3.

**Difficulty Assessment** To ensure stable RL training with consistent reward signals, we tag each problem's difficulty level, enabling a data mixture strategy that prevents all-zero rewards which could lead to training collapse. For each problem, we test DeepSeek-R1 four times, using the number of correct answers to determine difficulty. The formal definition appears in Appendix D.4, with the statistics of the dataset presented in Table 7 and tagging workflow in Appendix E.6.

**Test Set Annotation** Unlike the training set which used LLM labeling, our test set was manually annotated by 5 human experts using an open-web search engine. From 500 seed samples, experts followed the principles in Appendix D.2 to ensure meaningful evaluation of LLMs' information-seeking behaviors. Through iterative annotation, we finalized 275 samples for testing.

## 3.2 DeepDiver

Building upon the WebPuzzle, we showcase its efficacy within a RL framework designed to explore the information-seeking behavior of LLMs. In this section, we present our method, DeepDiver. DeepDiver ultilize the procedure of cold-start supervised fine-tuning (SFT) followed by reinforcement learning (RL), while incorporates a carefully designed reward assignment and scheduling mechanism to maintain stable RL training.

**Initialization of Reasoning and Searching** To equip DeepDiver with essential reasoning and searching capabilities for WebPuzzle, we implement a cold-start supervised fine-tuning process using diverse data: 2,000 WebPuzzle samples across difficulty levels, 300 real-user questions from our deployed smart assistant, 2,200 general reasoning problems from recent studies [10, 17, 35, 37, 43], and 1,000 real-user queries concatenated with retrieved documents. This dataset distills responses from DeepSeek-R1, establishing DeepDiver's foundational abilities to iteratively search and reasoning over retrieved documents. The distillation prompt configuration is detailed in Appendix E.6.

**GRPO With Iterative RAG** After SFT, we enhance DeepDiver by extending GRPO [24] with iterative RAG. As shown in Figure 3, the model iteratively performs reasoning and searching until reaching an acceptable answer, following the pipeline in Section 2.1. We apply a loss mask to distinguish model-generated from externally retrieved tokens, with GRPO updating parameters based solely on model-generated content.

**Extra Search Call Rewards** Beyond standard format and accuracy rewards in GRPO, we introduce an extra reward to encourage search engine use for complex problems. When **no search-free rollouts** solve a problem but **at least one search-enabled** rollout succeeds, we assign an additional reward of

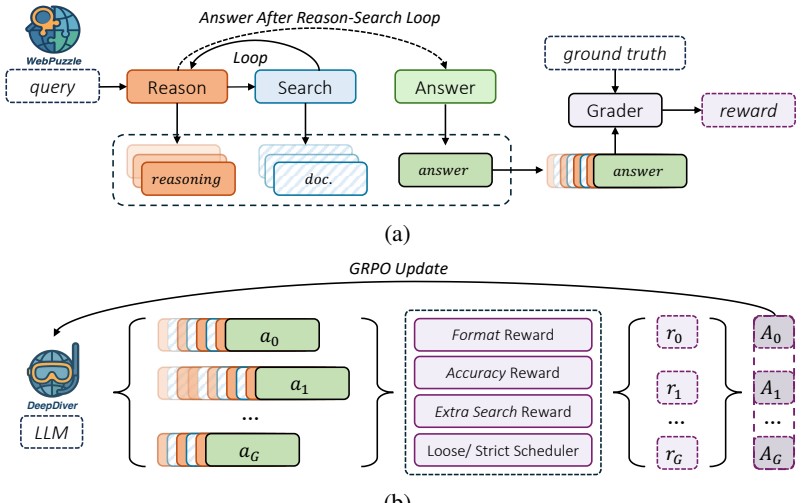

(a)

(b)

Figure 3: DeepDiver overview. (a) *Rollout Generation:* DeepDiver iteratively reasons, retrieves evidence, and answers WebPuzzle queries, then receives rewards based on comparison with ground truth. (b) *RL Updates:* Retrieved text is masked during loss calculation, and the LLM is refined via GRPO using advantages $A_i$ derived from rewards $r_i$.

1.0 to the successful search-enabled solutions. This ensures the model learns to leverage external tools when necessary. The formal definition appears in Appendix E.3.

**Loose and Strict Rewards** Our reward function employs LLM-based graders in a two-stage training approach that transitions from loose to strict grading: the loose grader assigns scores from 1 to 10 (scores ≥ 6 yield 1.0 reward), particularly benefiting early training as shown in Section 5.4. The strict grader conducts three evaluation rounds, requiring at least 2 of 3 positive judgments. Both grader definitions appear in Appendix E.2.

# 4 Experiments

## 4.1 Setup

**Data Mixture and Selection** Due to computational constraints and capability limits of the 7B model, we train DeepDiver on a carefully selected mixture of 7k WebPuzzle samples rather than the full dataset. We evenly split these into 2k samples for cold-start SFT (Section 3.2) and 5k for RL training. This mixture strategy balances computational efficiency and model effectiveness. Detailed statistics appear in Table 8.

**Benchmark Datasets and Baseline Models** We evaluate performance using closed-ended Chinese benchmarks including **C-simpleQA-500** [33, 8], **FRAMES-zh-230** [14], **BamBoogle-zh-71** [21], and our proposed **WebPuzzle** (detailed in Appendix E.4). For trainable baselines, we use Qwen2.5-7B-Instruct [30] and Pangu-7B-Reasoner [28] as backbone models. Training-free baselines include QwQ-32B [31], GPT-4o [20] and DeepSeek-R1 [6]. We evaluate methods including **Prompted without Web Search**, **Prompted with Iterative RAG**, and **R1-Distillation** (detailed in Appendix E.5). Our evaluation uses the strict grader from Section 3.2, which considers both reference answers and checklists for more robust assessment than conventional LLM-as-a-judge [44] approaches. Grader details appear in Appendix E.2.

## 4.2 How does DeepDiver's Performance Compare to Baselines?

*Our proposed DeepDiver demonstrates substantial improvements over distillation-based methods and achieves performance comparable to state-of-the-art models such as DeepSeek-R1 and QwQ.* As shown in Table 1, Qwen powered by DeepDiver achieves a 10-point improvement over the cold-start model on WebPuzzle, reaching 37.6 accuracy. DeepDiver-Qwen2.5-7B also outperforms the R1-distilled model (37.6 versus 29.8), highlighting the effectiveness of our RL training pipeline.

| | Open-Web Problems | Wiki-based Problems | | |
|---|---|---|---|---|
| | **WebPuzzle** | **C-SimpleQA-500** | **FRAMES-zh-230** | **BamBoogle-zh-71** |
| *Prompted without Web Search (Training-free)* | | | | |
| Qwen2.5-7B-Ins. [30] | 7.4 | 28.4 | 14.1 | 19.7 |
| Pangu-7B-Reasoner [28] | 15.0 | 36.3 | 20.4 | 27.2 |
| GPT-4o [20] | 14.2 | 61.8 | 51.7 | 52.6 |
| QwQ-32B [31] | 21.9 | 51.3 | 36.5 | 54.5 |
| DeepSeek-R1 [6] | **32.7** | **74.6** | **63.8** | **73.2** |
| *Prompted with Iterative RAG (Training-free)* | | | | |
| Qwen2.5-7B-Ins. | 17.0 | 65.3 | 30.9 | 40.8 |
| | (2.24) | (1.96) | (2.74) | (2.13) |
| Pangu-7B-Reasoner | 20.4 | 61.7 | 30.9 | 45.5 |
| | (3.80) | (1.87) | (2.09) | (2.41) |
| GPT-4o | 27.1 | 81.0 | 58.6 | 71.4 |
| | (1.39) | (1.24) | (1.56) | (1.29) |
| QwQ-32B | 31.4 | 79.0 | 50.4 | 73.2 |
| | (0.95) | (0.94) | (0.98) | (0.88) |
| DeepSeek-R1 | **37.1** | **84.8** | **65.8** | **79.3** |
| | (1.48) | (1.17) | (1.31) | (1.23) |
| *Training with Qwen2.5-7B-Insturct Series* | | | | |
| Cold- Start-SFT | 27.9 | 75.5 | 35.1 | 48.4 |
| | (1.85) | (1.35) | (1.73) | (1.24) |
| R1-Distill | 29.8 | 78.7 | 40.1 | 52.6 |
| | (1.75) | (1.32) | (1.56) | (1.34) |
| DeepDiver-Qwen2.5-7B | **37.6** | **81.9** | **44.5** | **63.4** |
| | (2.51) | (1.90) | (2.57) | (2.07) |
| *Training with Pangu-7B-Reasoner Series* | | | | |
| Cold-Start-SFT | 30.3 | 78.1 | 38.4 | 59.2 |
| | (1.84) | (1.37) | (1.84) | (1.49) |
| R1-Distill | 30.7 | 80.0 | 41.7 | 53.5 |
| | (1.77) | (1.34) | (1.83) | (1.40) |
| DeepDiver-Pangu-7B | **38.1** | **83.7** | **52.3** | **69.5** |
| | (2.89) | (2.61) | (3.05) | (2.72) |

Table 1: Numbers in () indicate average search call rounds per example, with each round potentially involving 1~5 search queries. Results show average accuracy across 3 runs.

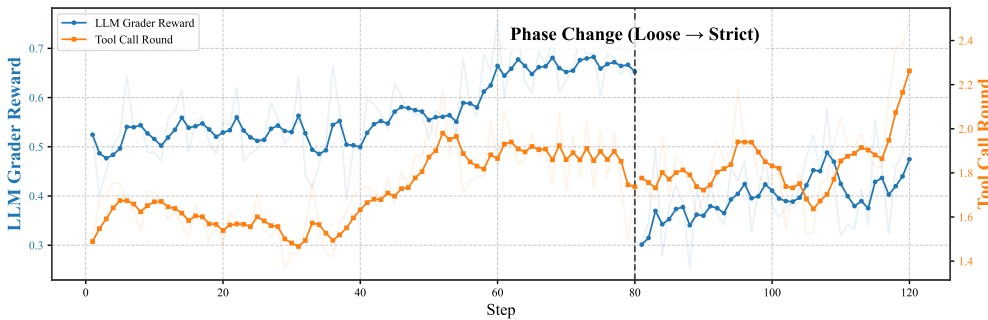

Figure 4: Correlation between reward value and the number of search calls across training phases. The increase in the number of search engine calls is accompanied by a rise in training rewards.

DeepDiver-Pangu-7B shows similar improvements. While R1-distilled Pangu-7B quickly hits performance bottlenecks (dropping 5.7 points on Bamboggle compared to the cold-start model), DeepDiver-Pangu-7B breaks through these limitations, showing substantial improvements across all benchmarks and achieving exceptional performance (38.1) on WebPuzzle. In conclusion, both Pangu-7B-reasoner and Qwen DeepDiver demonstrate competitive performance against high-performing models like DeepSeek-R1 and QwQ. This highlights DeepDiver's capability to effectively search for and reason over relevant information, and solve complex reasoning tasks through search intensity scaling.

### 4.3 What is the Relationship between Search Intensity and Performance?

*Search intensity is strongly correlated with performance improvements, as increases in search frequency and depth during the RL phase consistently lead to better outcomes.* Figure 4 illustrates this relationship during the training phase, showing a clear trend: as search engine calls increase, so do training rewards. For testing results, despite SFT's progress compared with the prompting methods,

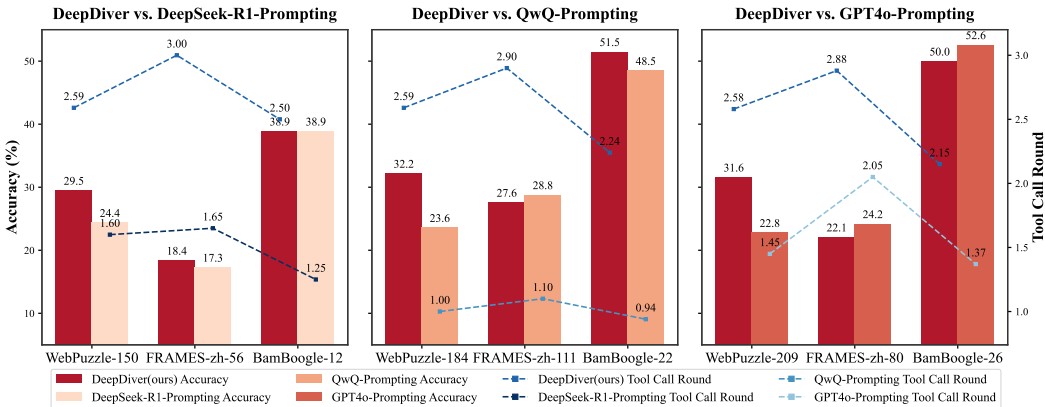

Figure 5: The comparison after removing cases answered correctly through internal knowledge.

the model faces a performance bottleneck to adapt to more challenging problems, still lagging behind off-the-shelf APIs with large margin. In contrast, our RL-based DeepDiver-Qwen2.5-7B promotes higher search intensity with an average of 2.51 search and reasoning rounds, substantially higher than the SFT model's 1.75. Similar gains appear in DeepDiver-Pangu-7B, where increased search rounds $(1.84 \rightarrow 2.89)$ correspond to performance improvements $(30.3 \rightarrow 38.1)$. This searching intensity scaling enables models to explore and verify more relevant information, enhancing their ability to tackle complex problems.

### 4.4 Can DeepDiver Generalize from Open-web Training to OOD Wiki-based Problem?

*Training with WebPuzzle, DeepDiver demonstrates strong generalization capabilities and performance improvements on Wiki-based problems.* DeepDiver shows impressive generalization on Wiki-based benchmarks despite not being specifically trained for these tasks. Both DeepDiver-Qwen2.5-7B and DeepDiver-Pangu-7B significantly outperform their distilled variants and demonstrate substantial improvements over cold-start models. While DeepSeek-R1 performs well on Wiki-based problems without web search, it shows modest gains when combined with iterative RAG pipeline. This suggests DeepSeek-R1 may have already internalized the necessary knowledge for Wiki-based problems, highlighting the importance of our proposed WebPuzzle benchmark. We further investigate this hypothesis through isolated tests on information seeking and verification in section 5.1.

## 5 Analysis

This section focuses on the Qwen2.5-7B-Instruct model, a simpler model comparing with the Pangu-7B-Reasoner. We analyze several key aspects, including isolated evaluations of information-seeking behavior, comparisons with concurrent related work, the design of the reward function, and the model's generalization to open-ended problems. Additional analyses—such as the **relationship between search intensity and problem difficulty, statistics of information-seeking behavior across different training and testing environments, comparisons between human and DeepDiver performance, and detailed case studies**—are provided in Appendix A.

### 5.1 Isolation Testing of Information-Seeking

While DeepDiver lags behind models such as QwQ and DeepSeek-R1 on certain datasets in Section 4, our primary focus is investigating information-seeking behavior rather than knowledge memorization. This raises a question: When isolating evaluation to focus purely on information seeking ability, how does DeepDiver compare to strong baselines?

**Setup** We conduct pairwise comparisons between DeepDiver and each baseline. For each pair, we perform $k = 3$ tests **without web search** to evaluate whether problems can be solved using internal knowledge alone. We calculate the $pass@k$ rate to filter out problems solvable by both models, then analyze accuracy on the remaining problems with the iterative RAG pipeline.

**Results** *DeepDiver exhibits exceptional information-seeking capabilities, comparable to all baselines on problems that cannot be solved by internal knowledge alone.* While our 7B DeepDiver initially trails behind 671B baselines in full-set tests, results shift when isolating information-seeking behavior. As Figure 5 shows, on problems challenging even for larger models, DeepDiver demonstrates competitive performance across all benchmarks. Notably, it outperforms DeepSeek-R1 across all domains, with a 5.1-point lead on WebPuzzle. This suggests our 7B model's limitations in full-dataset performance stem primarily from its smaller size limiting internal knowledge. However, when tackling problems requiring external information search and verification, DeepDiver's information-seeking capability demonstrates strength in addressing real-world open-web problems.

## 5.2 Comparisons with Wiki-based Methods

To highlight wiki-based training environments' limitations, we compare DeepDiver with prior wiki-based methods. Despite being trained entirely in Chinese, we evaluate DeepDiver on English benchmarks with English search engines to demonstrate its robustness and generalizability.

| | *Open-Web Problems* | *Wiki-based Problems* | | |
|---|---|---|---|---|
| | **WebPuzzle-en** | **BamBoogle** | **FRAMES** | **HotpotQA** |
| R1-Searcher [25] | 13.7 (1.9) | 46.7 (2.0) | 25.3 (1.9) | 57.9 (2.3) |
| DeepResearcher [45] | 15.0 (7.5) | 53.9 (7.1) | **33.6** (7.2) | 56.6 (4.4) |
| DeepDiver-Qwen | **26.1** (14.7) | **56.8** (9.1) | 32.0 (14.2) | **58.4** (10.4) |

Table 2: The comparison results with relevant works on the English evaluation dataset using English search engine environment. The number in () indicates the average number of search queries invoked.

**Setup** We use R1-Searcher [25] and DeepResearcher [45] as baselines—both trained in English using Wiki-based corpora. Search engine settings appear in Appendix E.8. For evaluation, we translate WebPuzzle into English via Qwen Max [30], use the full Bamboogle dataset [21] (125 examples), and randomly sample 300 examples from FRAMES [14] and HotpotQA [39]. For fairness, we report accuracy based on the average judgment across all methods.

**Results** *Despite the language gap, DeepDiver–trained in a real-world Chinese internet setting using WebPuzzle queries–outperforms Wiki-based baselines on most tasks, underscoring the strength of SIS.* As shown in Table 2, DeepDiver significantly outperforms DeepResearchers on WebPuzzle-en with 11.1 point leads, while maintaining strong results on Wiki-based datasets despite no English training for information-seeking. We attribute this to DeepDiver's use of SIS, which enables intensive information retrieval and verification rather than relying on limited internal knowledge or language-specific constraints. R1-Searcher and DeepResearcher make significantly fewer search calls than DeepDiver due to their "cleaner" and more constrained training environments, leading to poorer real-world performance when facing the noise and complexity of open information-seeking tasks. For additional results, including individual judge assessments, see Table 9.

## 5.3 Emergence of the SIS

A natural concern is that SIS could be an artifact of reward shaping rather than a genuine behavior that emerges from training in a real web environment. To address this concern, we analyze whether the extra search-call reward introduced in Section E.3 consistently encourages the model to prefer search over no-search when both solve the task.

**Setup** Theoretically, recall the bonus is only awarded when at least one search-enabled rollout succeeds and no search-free rollout succeeds for the same prompt group, i.e., it should **not** reward search when both search and no-search solve the task. Consequently, during training, we tracked the frequency of the extra bonus. Every 10 steps (448 trajectories per step), we counted how often the bonus fired and compared this against the evolving search intensity (average number of tool-use rounds per query).

**Results** *SIS is not merely reward-shaped but an emergent behavior developed during RL training, our reward design serves as a transient early scaffold (not persistent incentive) and differs fundamentally from existing works that explicitly encourages the tool calls.* As shown in Figure 6, the

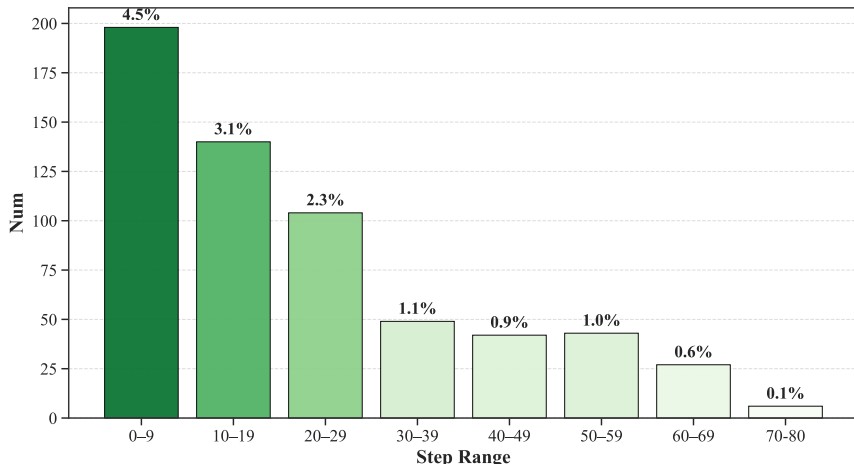

Figure 6: The count and percentage of trajectories triggering the special auxiliary search reward (value=3.0) across RL training step ranges. The trigger frequency drops sharply (4.5% in 0–9 steps to 0.1% in 70–80 steps) and fades after step 30, confirming the reward's transient role and supporting SIS as an emergent behavior.

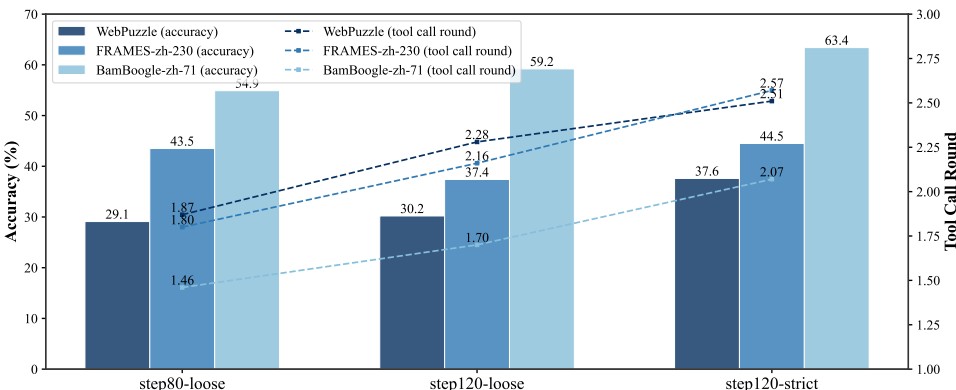

Figure 7: Training results with different reward functions show that a looser reward function stabilizes initial RL training, while a stricter reward function helps overcome later bottlenecks.

special search reward exhibits a clear phasing-out trend: its trigger percentage drops from 4.5% (198 occurrences) in steps 0–9 to 0.1% (6 occurrences) in steps 70–80. Two key observations confirm its transient role: (1) Even in the earliest training phase, only a small fraction of trajectories (4.5%) received the reward, ruling out "over-rewarding"; (2) After step 30, the trigger percentage remains ≤ 1.1%, indicating the reward is essentially inactive in mid-to-late training. More importantly, Figure 4 shows that tool-use rounds grow sharply during steps 80–120—well after the auxiliary reward fades. This confirms SIS is not a reward-driven "phenomenon" but an emergent behavior: the model proactively leverages external tools to compensate for internal knowledge limitations, even without direct incentives.

## 5.4 Tolerance of the Reward Function

During DeepDiver-Qwen7B's RL training, we observed a reward plateau after approximately 80 optimization steps. We investigated potential factors including learning rate scheduler, exploration diversity, environmental instability, and gradient issues, but found no obvious problems. We therefore focused on the reward function design as a potential cause of the performance plateau.

**Setup** Starting from checkpoints obtained after 80 optimization steps, we compared DeepDiver's performance under continued training with two different reward functions: the loose and strict

rewards introduced in Section 3.2. Both guided continued training from steps 80 to 120. We evaluated performance on WebPuzzle test sets, analyzing accuracy and search intensity trends.

**Results** *A looser reward function stabilizes the initial training phase of RL, while a stricter reward function helps overcome bottlenecks in the later stages.* Our results show that a looser reward function stabilizes early RL training, but continuing with it doesn't always lead to improvements. As Figure 7 shows, when transitioning from loose rewards (first 80 steps) to stricter rewards, we observed a nearly 9-point performance increase on WebPuzzle (from 29.1 to 37.6), compared to almost no improvement when continuing with loose rewards. On FRAMES-zh-230, continued training with loose rewards caused a sharp 7-point performance drop, while the stricter reward function continued driving performance upward.

### 5.5 Generalization to Open-ended Problems

DeepDiver is trained exclusively on closed-ended WebPuzzle problems, adaptively scaling search intensity based on complexity. We investigate whether these capabilities can generalize to open-ended tasks like long-form writing.

**Setup** We evaluate DeepDiver on ProxyQA [27] against R1-Distilled baselines. Since DeepDiver generates Chinese responses, we translate all ProxyQA meta-questions and sub-questions for evaluation. Testing prompt and evaluator configuration follow the original study. We analyze generalization benefits gained through RL training compared to distillation.

**Results** *RL training significantly enhances the generalization capability of LLMs, enabling transition from closed-ended to open-ended problems and demonstrating strong adaptability to long-form writing tasks.* As shown in Table 6, our RL-guided DeepDiver achieves $32.72\%$, outperforming the R1-distilled model by $9.47$ percentage points. This suggests RL training enables more effective information seeking and validation in open-web environments, resulting in more comprehensive responses. Additionally, DeepDiver's response length and search queries are substantially higher than the distilled model's, providing evidence that search intensity scaling encourages active information acquisition for more comprehensive answers.

## 6 Related Work

Prompting-based strategies—including in-context learning [3] and retrieval-augmented chain-of-thought pipelines [11, 32, 23]—enable zero- or few-shot question answering, yet their fixed templates rarely adapt retrieval depth to unforeseen information gaps. Supervised fine-tuning (SFT) improves the synergy between retrieval and generation [2, 41] but can overfit corpus-specific inference patterns, hindering transfer to noisy settings. Reinforcement-learning (RL) methods let LLMs decide *when* and *what* to search, achieving state-of-the-art results on curated benchmarks such as HotpotQA [12, 25, 4, 45, 39], yet they remain evaluated mostly in "clean" Wikipedia-style environments. Beyond these directions, tool-augmented agents that interleave reasoning with web search [18, 40, 38] similarly demonstrate promise but still rely on limited test beds, underscoring the need for benchmarks that reflect real-world, noisy information-seeking scenarios. Additional introduction of the related works are shown in Appendix B.

## 7 Conclusion

We conducted a comprehensive investigation into various aspects of information-seeking behavior in LLMs for solving real-world, knowledge-intensive problems. Our findings indicate that an RL-driven framework, when combined with open-web search engines, enables LLMs to scale search intensity and adapt to tasks of varying difficulty levels. We introduced WebPuzzle, a large-scale dataset designed specifically for developing and testing LLMs' information-seeking behavior, and DeepDiver, a 7B parameter LLM powered by WebPuzzle, which demonstrates competitive performance when compared to the 671B DeepSeek-R1 model on knowledge-intensive tasks. Additionally, we explored key factors influencing RL training and the behavior of LLMs. Our work empowers LLMs to spontaneously adapt their seeking behavior, contributing to advancements in the field and providing extensive insights into the information-seeking capabilities of LLMs in real-world tasks.

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

# Appendix

## A  Further Analysis

### A.1  Search Intensity *vs.* Difficulty

To showcase DeepDiver's ability to dynamically adjust search intensity based on problem complexity, we examine the relationship between search intensity and accuracy across fractions with varying difficulty levels in the WebPuzzle. This analysis follows the experimental setup outlined in Section 4 and compares our proposed DeepDiver with the DeepSeek-R1 baseline.

| Methods | WebPuzzle | | | |
| --- | --- | --- | --- | --- |
| | Cross-Page QA-130 | Open&Wiki Reddle-145 | Easy&Medium-96 | Hard&Outliers-179 |
| DeepSeek-R1 (w/o search) | 32.6 (0.00) | 32.9 (0.00) | 53.5 (0.00) | 21.6 (0.00) |
| DeepSeek-R1 Iterative RAG | 43.8 (1.31) | 31.0 (1.64) | 61.1 (1.30) | 24.2 (1.58) |
| Qwen7b-Ins-R1-Distill | 37.2 (1.49) | 23.2 (1.99) | 45.2 (1.62) | 21.6 (1.83) |
| DeepDiver (Ours) | 47.4 (2.35) | 28.8 (2.65) | 55.6 (2.34) | 27.9 (2.60) |

Table 3: The performance of different subsets in WebPuzzle. The number in () indicates the average number of search call rounds on the subset.

**Results**  *DeepDiver demonstrates significant benefits from adaptive search intensity scaling, where search intensity is proportional to both problem difficulty and the LLM's performance.*  As shown in table 3, across all difficulty levels, both DeepDiver and the baseline models show an increasing number of search call rounds as problem complexity rises. However, DeepDiver consistently consumes more search calls, which translates into better performance. In particular, when compared to DeepSeek-R1, DeepDiver outperforms it in the hard and outlier fractions by a large margin. Specifically, DeepDiver achieves a notable 3.7-point performance leading, driven by an average of 2.6 search rounds compared to DeepSeek-R1's 1.59. This demonstrates that DeepDiver, empowered by open-web environment and reinforcement learning, exhibits superior performance on more complex problems.

An interesting observation arises when examining the performance of DeepSeek-R1 on the Wiki Riddle fraction. Although equipped with the iterative RAG pipeline, DeepSeek-R1 experiences a 1.9-point performance drop (from 32.9 to 31.0). We hypothesize that this decline is due to knowledge conflicts between the pre-trained internalized Wiki corpus and the real-world open-web environment, which introduces confusion and hallucination that hinders the model's ability to correctly answer the question. These results further validate the effectiveness of the WebPuzzle and DeepDiver.

### A.2  Statistics of Information-Seeking Behaviors

To further analyze the differences between our WebPuzzle dataset and wiki-based datasets, as well as to conduct an in-depth investigation into the behavior of our DeepDiver model, we computed the proportions of different information-seeking behaviors (defined in section 2.2) across various models on multiple datasets. The detailed results are presented in Table 4.

| | Methods | WebPuzzle-en | BamBoogle | FRAMES | HotpotQA |
| --- | --- | --- | --- | --- | --- |
| | R1-Searcher | 0.04 | 0.03 | 0.04 | 0.02 |
| Reflection & Correction | DeepResearcher | 0.20 | 0.11 | 0.13 | 0.07 |
| | DeepDiver-Qwen | **0.45** | **0.25** | **0.28** | **0.27** |
| | R1-Searcher | 0.06 | 0.02 | 0.02 | 0.02 |
| Conflict Resolution | DeepResearcher | 0.16 | 0.07 | 0.08 | 0.06 |
| | DeepDiver-Qwen | **0.31** | **0.23** | **0.32** | **0.18** |
| | R1-Searcher | 0.17 | 0.18 | 0.13 | 0.10 |
| Verification & Denoising | DeepResearcher | 0.32 | 0.15 | 0.24 | 0.18 |
| | DeepDiver-Qwen | **1.72** | **1.80** | **1.60** | **1.54** |

Table 4: Behavior statistics of multiple models on our WebPuzzle dataset and several wiki-based datasets. Each value in the table represents the average occurrence count of a rollout-level behavior.

**Setup**    We developed an automatic pipeline based on GPT-4o [20] to identify and count the occurrences of different behaviors in the reasoning chains of model outputs. We primarily focused on three behaviors: Reflection & Correction, Conflict Resolution, and Verification & Denoising, while omitting the statistics for Evidence Gathering & Supplements due to its overly fundamental nature. Detailed prompt specifications can be found in Figure 16. The baseline models, evaluation benchmarks, and analysis setup align with those in Experiment 5.2.

**Results**    *The WebPuzzle proves to be more challenging compared to wiki-based datasets, requiring more complex information-seeking behaviors. Additionally, DeepDiver's reasoning chains exhibit richer and more diverse patterns.* As shown in tabel 4, The behavioral statistics of WebPuzzle-en surpass those of wiki-based datasets across nearly all models, particularly in Reflection & Correction and Conflict Resolution behaviors. Notably, for the HotpotQA dataset - the primary training data for both baseline models - we observe a substantial reduction in the proportion of complex behaviors required compared to WebPuzzle when evaluating the same model, further highlighting the importance of more open-web training data. Additionally, DeepDiver, trained on WebPuzzle with real search engine integration, demonstrates richer and more sophisticated information-seeking behaviors compared to various baseline models.

### A.3    Human vs. DeepDiver

To assess the difficulty of WebPuzzles and compare human performance with that of DeepDiver, we conducted human testing on a subset of the WebPuzzle evaluation dataset. Human performance was evaluated by tracking the number of web-search queries and web pages browsed during the problem-solving process, which were then compared with the performance of DeepDiver.

**Setup**    Five human experts, who were not involved in the annotation of the evaluation sets, participated in the human evaluation. Each expert was tasked with answering 5 problems, creating a subset of the evaluation set. During testing, the number of search queries and web pages browsed were recorded. Consistent with the experimental setup, we report the average accuracy of the human evaluators and compare it against DeepSeek-R1 and our proposed DeepDiver to highlight the challenges posed by WebPuzzle.

|  | WebPuzzle | | | |
|---|---|---|---|---|
|  | *Accuracy* | *# Search Rounds* | *# Search Queries* | *# of Page Browsed* |
| Human | **44.0** | - | 6.16 | 9.28 |
| GPT-4o | 30.7 | 1.47 | 5.28 | - |
| QwQ-32B | 27.0 | 0.95 | 3.77 | - |
| R1-Distill | 38.7 | 1.81 | 6.79 | - |
| DeepSeek-R1 | 26.7 | 1.94 | 7.37 | - |
| DeepDiver | **40.0** | **2.68** | **10.69** | - |

Table 5: Human evaluation results for 25 randomly sampled questions from WebPuzzle evaluation dataset, compared with baseline models and DeepDiver.

**Results**    *WebPuzzle presents significant challenges, even for human experts, who face difficulties with numerous searches and reasoning steps.* As shown in Table 5, human evaluators achieved an accuracy rate of 44.0%, requiring an average of 6.16 search queries and browsing 9.28 web pages to solve the problems. In comparison, DeepSeek-R1 made 7.37 search queries with 1.94 rounds of searching. Despite DeepSeek-R1 performing the most search rounds among all baselines and humans, it achieved a relative lower accuracy of 38.7%. In contrast, our proposed DeepDiver conducted 10.69 search queries across 2.68 rounds, significantly surpassing the other methods in terms of search effort. This additional effort allowed DeepDiver to more thoroughly collect and verify evidence, leading to a 40.0% accuracy rate, which is closer to human performance. These results suggest that DeepDiver follows a more comprehensive information-seeking process, better aligning with the approach taken by human evaluators.

## A.4 Case Study

In this section, we conduct an error analysis and case study on the response generated from DeepSeek-R1, R1-distilled Qwen-7b and our DeepDiver. We explain the reason why the DeepDiver outperforms the R1-distilled model and show competitive performance compared with the DeepSeek-r1 using one typical example.

**Results** *DeepDiver demonstrates exceptional information-seeking ability while incorporating correction on reasoning history and retrieved documents, providing a more robust and adaptable solution for overcoming the limitations of flawed internal knowledge.* Specifically, as shown in Table 10, DeepSeek-R1 leverages its rich internal knowledge to quickly narrow the exploration scope, consistently demonstrating the ability to list the correct answer among candidate options in the first round, showcasing remarkable knowledge retention. Taking advantage of this, R1 can focus more on verifying whether candidate answers satisfy all constraints when designing search queries, allowing it to find the correct answer in fewer rounds. However, the R1-distilled Qwen-7B attempts to mimic DeepSeek-R1's behavior but lacks error correction when internal knowledge is flawed. Specifically, the R1-distilled Qwen suggests "Nico Hülkenberg" in the first round of searching and reasoning, but fails to resolve conflicting conditions in subsequent rounds due to limited internal knowledge, ultimately producing a faulty answer.

In contrast, without R1-level internal knowledge, DeepDiver compensates by increasing search intensity to acquire more relevant external documents. This results in 7 generated search queries across rounds 1 and 2 to explore diverse documents, rather than relying on limited internal knowledge. Furthermore, in rounds 2-3, DeepDiver encounters a potential candidate, "Chaz Mostert," but allocates only 1 query for validation, compared to 3 queries for continued exploration. This persistent exploration enables the model to identify the correct answer by round 3. The search and reasoning strategy, empowered by SIS, aids DeepDiver in delivering a correct and acceptable answer.

## A.5 Results on ProxyQA

We evaluate DeepDiver on ProxyQA [27] against R1-Distilled baselines. Since DeepDiver generates Chinese responses, we translate all ProxyQA meta-questions and sub-questions for evaluation. Testing prompt and evaluator configuration follow the original study. We analyze generalization benefits gained through RL training compared to distillation.

| | ProxyQA | | | |
|---|---|---|---|---|
| | *Acc.* | *# rounds* | *# searches* | *response length* |
| R1-Distill | 23.25% | 1.76 | 7.39 | 590.31 |
| DeepDiver | 32.72% | 2.27 | 10.54 | 1971.58 |

Table 6: Results on ProxyQA. The results include accuracy rate, number of search calls, rounds of search, the number of search queries, and response length.

# B Additional Related Works

**Prompting-based strategies.** Few-shot in-context learning [3] allows frozen LLMs to imitate reasoning patterns from exemplars, while RAG with CoT prompts [11, 32, 23] interleave search queries with intermediate thoughts to inject fresh evidence. These methods require no training, but their step counts and query formats are anchored to the prompt, so the model cannot escalate effort when initial evidence is missing. As a result, they often stop searching too early or hallucinate unsupported facts on hard, open-web questions.

**Supervised fine-tuning (SFT).** Retrieval-augmented generation systems fine-tuned on gold passages—e.g., Self-RAG and Auto-RAG [2, 41]—learn to quote and merge the external text into answers, reducing hallucinations on Wikipedia benchmarks. Nevertheless, SFT tends to overfit the inference paradigm tied to the training corpus; performance drops when pages are noisy, multilingual, or partially missing, which is common in open-web internet environment.

**Reinforcement learning (RL).** RL integrated iterative RAG pipeline offers a principled way for LLM agents to decide *when* and *what* to search. Recent work trains search-capable agents with reward shaping and curricula, achieving state-of-the-art accuracy on HotpotQA and similar wiki-based corpus [12, 25, 4, 45]. Yet these studies use closed Wikipedia environments, where every answer is guaranteed to exist and pages are clean.

**Tool-augmented agents.** ReAct-style frameworks combine chain-of-thought with executable actions, letting models call a browser or other tools between reasoning steps [18, 40, 38]. They excel at citing fresh evidence and correcting themselves mid-trajectory, but they still struggle with irrelevant pages and and hard to dealing with the real-world open-web environment.

**Group Relative Policy Optimization** Group Relative Policy Optimization (GRPO) [24] is a reinforcement learning algorithm designed to enhance the efficiency of Proximal Policy Optimization (PPO) [22] by eliminating the need for critic network. Specifically, GRPO samples multiple outputs from a previous policy for a given prompt and computes their average reward to serve as a dynamic baseline. The advantage of each output is defined relative to this baseline, resulting in positive advantages for outputs exceeding the baseline and negative advantages otherwise. Formally, we define the GRPO objective as follow:

$$\mathcal{J}_{\text{GRPO}}(\theta) = \mathbb{E}\left[\sum_{i=1}^{G} \min\left(\frac{\pi_\theta(o_i)}{\pi_{\theta_{\text{old}}}(o_i)} A_i,\ \text{clip}\left(\frac{\pi_\theta(o_i)}{\pi_{\theta_{\text{old}}}(o_i)}, 1 - \epsilon, 1 + \epsilon\right) A_i\right) - \beta\, \mathbb{D}_{\text{KL}}\left(\pi_\theta \,\|\, \pi_{\text{ref}}\right)\right], \quad (1)$$

where the relative advantage $A_i$ is computed as:

$$A_i = \frac{r_i - \text{mean}(\{r_1, r_2, \ldots, r_G\})}{\text{std}(\{r_1, r_2, \ldots, r_G\})}. \quad (2)$$

GRPO retains PPO's clipping strategy and incorporates a KL-divergence term for regularization to ensure the stable training of RL.

## C  Discussion: Limitations and Extensions

In this work, we aimed to provide a deeper understanding of the information-seeking behavior of large language models (LLMs) with respect to real-world open-web challenges. While we observed that DeepDiver outperforms multiple baselines in several cases, we cannot claim that DeepDiver is the most optimized solution for enabling LLMs to solve complex problems in a broad sense. Below, we outline several limitations of our work and potential areas for future exploration.

**Curation of WebPuzzle** We argue that DeepSeek-R1 excels at solving problems by leveraging internal knowledge rather than extensively exploring the internet. As a result, we designed the WebPuzzle to assess and test the LLM's information-seeking abilities. However, over time, there is a possibility that cutting-edge LLMs will internalize the knowledge sources used in WebPuzzle, which could lead to scenarios similar to those seen in Wiki-based problems. Moreover, the curation pipeline heavily relies on the utilization of DeepSeek-R1, introducing some potential bias in the testing process. Therefore, developing effective benchmarks to assess the evolving capabilities of LLMs will remain an ongoing challenge.

**RL-Driven Open-ended Problems Solving** We demonstrate that DeepDiver, trained on closed-ended open-web problems, exhibits strong generalization capabilities when applied to open-ended tasks, such as long-form writing. However, developing a more effective RL-driven framework that directly enhances the model's ability to solve these problems remains an open challenge. Since the RL framework relies on a stable reward signal, the lack of reliable metrics to assess open-ended content complicates the task of defining such signals. As a result, designing a framework that can handle both open-ended and closed-ended problems remains a key area for future exploration. One potential approach could following the approaches of ProxyQA, involving crafting sub-questions in an online way during RL training, which may offer a promising direction for enhancing the model's performance on open-ended tasks.

**The Border Between Cold Start SFT and RL**   DeepDiver is powered by a cold-start SFT and RL pipeline, which first initializes the model's capabilities through SFT training, followed by RL training. However, there is no established guideline on the optimal extent to which SFT should be conducted before transitioning to RL training. This presents a challenge, as maintaining stable and effective RL training requires researchers to experiment with various proportions and combinations of SFT samples, while continuously monitoring the RL training process. Future improvements could involve an adaptive pipeline, allowing the LLM to dynamically switch from SFT to RL training when necessary.

**The Extension of Tool Usage**   Search engines are commonly viewed as tools that LLMs can utilize to enhance their capabilities. In our study, we focused solely on investigating the reasoning and searching behavior of LLMs with the aid of search engines. This limits the scope of our work, as there are various other tools, such as those compatible with the Model Context Protocol (MCP) [1], which could also contribute to improving reasoning and searching processes in knowledge-intensive tasks. Future research could expand on this by considering the integration of additional tools to further enhance the performance of LLMs in such tasks.

**Scalability with Respect to Model Size and Sequence Length**   Due to computational constraints, the experiments presented in this report are limited to a 7B model with a maximum sequence length of 20k tokens. This limitation restricts the generalizability of our findings to larger models (e.g., 13B, 32B) and longer sequence lengths (e.g., 32k, 64k), and it also limits our exploration of DeepDiver's upper performance boundaries. Currently, DeepDiver's performance is predominantly constrained by both model size and response length. Extending the training and evaluation to encompass these configurations in future work would provide valuable insights into the model's scaling behavior. Such an extension would help assess DeepDiver's performance, robustness, and applicability across different model sizes and sequence lengths, offering a more comprehensive understanding of its real-world potential.

**Problem of Over-searching**   Prior work has shown that reasoning LLMs trained in RL environments often suffer from "overthinking" [5], generating excessively long reasoning sequences—sometimes spanning thousands of tokens—even for simple questions. We observe a similar phenomenon in our DeepDiver model, which invokes significantly more search calls compared to other baselines, even when evaluated on simple tasks. This over-searching behavior highlights an inefficiency that future work should aim to mitigate. A promising direction would be to develop methods that reduce both the search and reasoning overhead in LLM-based systems.

# D   Detailed Dataset Curation

## D.1   Examples of the WebPuzzle

In this section, we show two examples of the WebPuzzle sampled from the cross-page question and riddles respectively.

---

**Example of Cross-page Question**

2024年10月31日，一名身高125cm的普通成人游客购买深圳欢乐谷万圣节夜场票，需支付多少钱？

*Translation: On October 31st, 2024, how much does an ordinary adult tourist with a height of 125 cm need to pay for the night ticket of the Halloween event at Happy Valley Shenzhen?*

---

**Solution: 149 RMB**

某中国社交媒体应用因美国政府的禁令而意外获得大量关注，用户自称为"难民"并迁移到该平台。该应用在苹果应用商店一度下载量排名第一，超过ChatGPT。其母公司总部位于上海。这是哪个应用？

*Translation: A Chinese social media app unexpectedly gained a great deal of attention due to a U.S. government ban. Users refer to themselves as "refugees" and have migrated to this platform. At one point, the app ranked first in downloads on the Apple App Store, surpassing ChatGPT. Its parent company is headquartered in Shanghai. Which app is this?*

## D.2 Principles to Annotate the Evaluation Set

The experts were required to adhere to the following principles: (1) The answer to the posed question must be definitive and unique. (2) The answer should be derived from internet search results, not from common-sense knowledge. (3) Answering the question should involve multiple searches, thorough reasoning, and validation, rather than a simple query. (4) The answer must be accessible and solvable through sufficient rounds of searching and reasoning.

## D.3 Quality Assurance of WebPuzzle

We prioritize recently updated pages not fully covered by **LLMs' knowledge cutoffs**, emphasizing open-web searches for problem-solving. We exclude offensive, politically sensitive, ethically concerning, or NSFW content using an LLM-based filter. Ambiguous, controversial, multiple-choice and boolean questions are removed to prevent answer hacking. We also eliminated unsolvable problems to ensure low error rates, ultimately collecting 24k WebPuzzle training samples.

## D.4 Diffculty Level Tagging

Formally, let $N_{\text{correct}}$ denote the number of correct answers out of the 4 tests. The difficulty level $D$ of a problem can be determined as:

$$D = \begin{cases} \text{easy} & \text{if } N_{\text{correct}} = 4 \\ \text{medium} & \text{if } N_{\text{correct}} = 2 \text{ or } 3 \\ \text{hard} & \text{if } N_{\text{correct}} = 1 \\ \text{outlier} & \text{if } N_{\text{correct}} = 0 \end{cases} \tag{3}$$

# E Detailed Experimental Setup

## E.1 Statistic of WebPuzzle

| Data Category | Training Data Num | | | | | Evaluation Data Num | | | | |
|---|---|---|---|---|---|---|---|---|---|---|
| | Easy | Medium | Hard | Outliers | ALL | Easy | Medium | Hard | Outliers | ALL |
| Cross-Page QA | 2553 | 2451 | 1404 | 3970 | 4500 | 5 | 54 | 27 | 44 | 130 |
| Open&Wiki Riddle | 5566 | 2956 | 1409 | 3375 | 2500 | 0 | 37 | 33 | 75 | 145 |
| Total Set | 8119 | 5407 | 2813 | 7345 | 23684 | 5 | 91 | 60 | 119 | 275 |

Table 7: Data statistics of the full WebPuzzle dataset. Problems in WebPuzzle are labeled as easy, medium, or hard, and outliers refer to cases with $pass@4 = 0$.

## E.2 Reward Defination

Formally, Let $\mathcal{G}_i$ denote the similarity score assigned by the looser grader for the $i$-th response. The reward $\mathcal{E}_i$ assigned by the looser grader is defined as:

$$\mathcal{E}_i = \begin{cases} 1.0 & \text{if } \mathcal{G}_i \geq 6, \\ 0.0 & \text{otherwise.} \end{cases} \tag{4}$$

For the stricter grader, the response undergoes three rounds of evaluations, each providing a binary judgment, $\mathcal{J}_i^k \in 0, 1$, where $k \in 1, 2, 3$ represents the corresponding round. The reward $\mathcal{E}_i$ is assigned only if at least two out of the three rounds agree that the response is semantically equivalent to the reference. The stricter grader's reward mechanism is defined as:

$$\mathcal{E}_i = \begin{cases} 1 & \text{if } \sum_{k=1}^{3} \mathcal{J}_i^k \geq 2, \\ 0 & \text{otherwise.} \end{cases} \tag{5}$$

the strict grader evaluates the generated response by comparing it to the reference answer and the checklists over three rounds. In each round, the evaluator determines whether the model's output matches the reference answer and aligns with the requirements specified in the checklists. If the evaluator deems the response correct in at least two out of the three rounds, the result is classified as correct; otherwise, it is considered incorrect. The accuracy rate, derived from this evaluation process, is reported as the primary metric in our results.

### E.3 Extra Tool Call Rewards Assignment

Formally, the triggering condition for extra rewards is defined as:

$$\forall i \in G, \mathcal{S}_i = 0 \implies \mathcal{C}_i = 0, \quad \exists j \in G \text{ such that } \mathcal{S}_j = 1 \text{ and } \mathcal{C}_j = 1, \tag{6}$$

where $G$ is the group of rollouts, $\mathcal{S}_i$ indicates whether the $i$-th rollout uses a search engine, and $\mathcal{C}_i$ indicates success. The extra reward is assigned only when no search-free rollouts succeed and at least one search-enabled rollout succeeds. In such cases, a +1.0 reward is added to successful search-enabled rollouts.

Formally, we apply the extra reward with:

$$\mathcal{E}_i = \begin{cases} 1.0 & \text{if } \mathcal{S}_i = 1 \text{ and } \mathcal{C}_i = 1, \\ 0.0 & \text{otherwise.} \end{cases} \tag{7}$$

### E.4 Benchmarks

We evaluate the performance of all models using the following benchmarks:

- **WebPuzzle**: Our proposed novel, web-based question-answering benchmark designed to assess models' deep information-seeking abilities within a real-world web environment. WebPuzzle serves as an in-domain task that evaluates a model's capacity to extract and process information from the web.

- **C-SimpleQA-500**: A randomly sampled subset of 500 instances from C-SimpleQA [33], C-SimpleQA is a Chinese-translated version designed to assess the factuality of language models. While not explicitly designed for complex or real-time question answering, we utilize C-SimpleQA to explore the impact of web search scaling on simpler information-seeking tasks.

- **FRAMES-zh-230**: A subset of the FRAMES [14] benchmark with 230 samples that requires multi-hop information-seeking. The queries are translated into Chinese, and annotators verify whether the golden answer can be retrieved via interactions with our web search API. Only test cases where the golden answer is reachable are included in the evaluation.

- **Bamboogle-zh-71**: A subset of the Bamboogle [21] benchmark consists of 71 samples, processed similarly to the FRAMES-zh-230 dataset.

### E.5 Baseline Methods

The following baseline methods are evaluated:

- **Prompted without Web Search**: In this setup, LLMs respond to problems based on a single round of prompting without web access. The model generates answers based solely on its pre-existing knowledge. We include off-the-shelf LLMs such as Qwen2.5-7B-Instruct [30, 36], QwQ-32B [31], GPT-4o [20], DeepSeek-R1 [6] and Pangu-7B-Reasoner [28].

- **Prompted with Iterative RAG**: In this approach, LLMs answer problems through multiple rounds of retrieval and reasoning, using a real-world open-web search engine. The baselines from the "Prompted without Web Search Methods" are tested using the same iterative RAG workflow in our approach (see Appendix E.6 for implementation details).

- **R1-Distillation**: We performed SFT on Qwen2.5-7B-Instruct and Pangu-7B-Reasoner using a combined dataset that integrates both the cold-start data (introduced in Section 3.2) and the dataset used during the RL training phase of DeepDiver. Both models are distilled using the responses generated by DeepSeek-R1.

## E.6 Iterative RAG Prompting

In our experimental framework, which includes WebPuzzle data tagging (Section 3.1), cold start data construction (Section 3.2), DeepSeek-R1 distillation, and multiple iterative RAG baselines (Section 4.1), we utilize prompt-based approaches to implement iterative RAG. We have specifically designed a prompt where, in each round, we evaluate the model's response to decide whether to terminate the process. If retrieval is triggered, the retrieved results are concatenated in the next user turn to facilitate continued reasoning and searching. For a detailed description of our prompt design, please refer to Figure 12.

## E.7 Grader Details

During the training of the RL model, both a loose grader and a strict grader are involved. The loose grader performs a single evaluation and provides a score ranging from 1 to 10, with a score of 6 or above considered correct. This loose grader is used in the early stages of training to enhance training signals and ensure stability. In contrast, the strict grader performs three rounds of validation using different prompts, each producing a binary classification of "correct" or "incorrect." The final result is determined by majority voting across the three validations. This grader is used in the later stages of training to prevent model manipulation and further improve performance. Both graders are implemented base on qwen-turbo API[*]. For details on the specific grader prompts, please refer to Figure 14 and 15.

## E.8 Implementation Details

During training, each data sample undergoes 14 rollouts with a sampling temperature of 0.9. We employ a batch size of 32 and a learning rate of 1e-6, training for a single epoch with a KL divergence coefficient of 0.001. The maximum number of tool call round is set to 7. For online search, we utilize the Bocha[†] search engine for Chinese scenario and LangSearch[‡] for English scenario, retaining only the top 2 results per search query to ensure efficiency.

| Data Category | Training Data Num | | | | | Evaluation Data Num | | | | |
|---|---|---|---|---|---|---|---|---|---|---|
| | Easy | Medium | Hard | Outliers | ALL | Easy | Medium | Hard | Outliers | ALL |
| Cross-Page QA | 200 | 2200 | 1300 | 800 | 4500 | 5 | 54 | 27 | 44 | 130 |
| Open&Wiki Riddle | 0 | 1200 | 1100 | 200 | 2500 | 0 | 37 | 33 | 75 | 145 |
| Total Set | 200 | 3400 | 2400 | 1000 | 7000 | 5 | 91 | 60 | 119 | 275 |

Table 8: Data statistics of the WebPuzzle training and evaluation sets uesd in our Experiment. Problems are labeled as easy, medium, or hard, and outliers refer to cases with $pass@4 = 0$.

---

[*]https://www.alibabacloud.com/help/en/model-studio/what-is-qwen-llm

[†]https://open.bochaai.com/

[‡]https://langsearch.com/

| | Methods | WebPuzzle-en | BamBoogle | FRAMES | HotpotQA |
|---|---|---|---|---|---|
| Our LLM Judge | R1-Searcher [25] | 6.5 | 44.8 | 24.0 | 56.7 |
| | DeepResearcher [45] | 5.5 | 51.2 | 32.3 | 53.7 |
| | DeepDiver-Qwen | 21.1 | 61.6 | 36.0 | 64.3 |
| R1-Searcher LLM Judge | R1-Searcher | 13.5 | 46.4 | 24.3 | 54.7 |
| | DeepResearcher | 15.5 | 51.6 | 32.8 | 54.8 |
| | DeepDiver-Qwen | 16.0 | 43.2 | 22.4 | 44.0 |
| DeepResearcher LLM Judge | R1-Searcher | 21.1 | 48.8 | 27.7 | 62.3 |
| | DeepResearcher | 24.0 | 58.9 | 35.8 | 61.2 |
| | DeepDiver-Qwen | 41.1 | 65.6 | 37.5 | 67.0 |
| Average | R1-Searcher | 13.7 (1.9) | 46.7 (2.0) | 25.3 (1.9) | 57.9 (2.3) |
| | DeepResearcher | 15.0 (7.5) | 53.9 (7.1) | **33.6** (7.2) | 56.6 (4.4) |
| | DeepDiver-Qwen | **26.1** (14.7) | **56.8** (9.1) | 32.0 (14.2) | **58.4** (10.4) |

Table 9: The full comparison results with relevant works on the English evaluation dataset using English search engine environment. We took the average of the LLM Judge's results from three works to evaluate the accuracy. The number in () indicates the average number of search queries invoked.

# F  Case Study Examples

In this section, we present a case study on the WebPuzzle, comparing the performance of our proposed DeepDiver model with the R1-distilled model and DeepSeek-R1. The results are shown in Table 10.

## F.1  ProxyQA Case

In this section, we present an example of DeepDiver's results on ProxyQA. We compare the performance of DeepDiver-Qwen2.5-7B with its R1-distilled variant. The original R1 model only requires a single round of reasoning due to its strong internal knowledge. However, as shown in Figure 9, the distilled model replicates the single round of reasoning without possessing the same internal knowledge. This lack of depth leads to overconfidence, causing the model to prematurely provide an answer after just one round of reasoning. In contrast, Figure 8 demonstrates that our proposed DeepDiver consistently searches and reasons over multiple rounds to gather the necessary knowledge required to answer the problem, resulting in a more thorough and comprehensive output.

| Problem | 他职业生涯中曾打破一级方程式最年轻获得积分车手的纪录，
先后效力过三支不同顶级车队的F1阵营，
并两度经历席位调换的特殊安排，曾于德国大奖赛助力某中游车队斩获2008年后的首个领奖台，
转战耐力赛领域后，他驾驶26号赛车在多项赛事中缔造了新的历史
这位车手究竟是谁？ |
| --- | --- |
| Solution | 丹尼尔·科维亚特 |
| R1-Prompting | **\<think\>**: ...维斯塔潘保持最年轻积分纪录...霍肯博格在2010年拿积分时年龄未破纪录...科维亚特（红牛二队属中游但耐力赛未驾驶26号车）...
**\<search\>**: ["F1最年轻获得积分车手纪录", "效力三支F1车队的车手德国大奖赛领奖台", "2019德国大奖赛领奖台车队", "WEC 26号车车手F1背景"]
**Search Results:** ...correct answer is retrieved by "2019德国大奖赛领奖台车队"

**\<think\>**: ...维斯塔潘排除...科维亚特需校验...亚历山大·阿本需确认是否存在其他车手关联。
**\<search\>**: ["科维亚特效力F1车队数量", "2019德国大奖赛红牛二队是否中游车队", "F1车手驾驶WEC 26号车", "丹尼尔·科维亚特耐力赛"]
**Search Results:** ...(correct answer involved)

**\<think\>**: ...科维亚特符合"2008年后中游车队首登德国站领奖台"条件...其他如霍肯博格阿尔本均不符合全部条件。
**\<answer\>**: ...丹尼尔·科维亚特✓... |
| Qwen7b-R1-Distill | **\<think\>**: ...初步判断可能为尼科·霍肯伯格，...
**\<search\>**: ["F1最年轻积分车手记录保持者", "尼科·霍肯伯格席位调换历史", "2014年德国大奖赛索伯纳尔逊·皮奎特", "勒芒24小时赛26号车霍肯伯格"]
**Search Results:** ...(correct answer is not included)

**\<think\>**: ...尼科·霍肯伯格从未登上过领奖台，这与德国站季军存在冲突...
**\<search\>**: ["2014年德国大奖赛索伯车队成绩", "F1最年轻积分车手记录保持者", "勒芒24小时赛26号车手霍肯伯格"]
**Search Results:** ...(correct answer is not included)

**\<think\>**: *矛盾点处理：F1历史数据显示2014年德国站中游车队（非顶级车队）的最好成绩为马萨（法拉利车队季军），而索伯车队该站最佳为霍肯伯格季军。此处可能存在用户将"中游车队"误解为"非顶级车队"的表述误差，需在回答中明确说明。*
**\<answer\>**: ...尼科·霍肯伯格✗... |
| DeepDiver | **\<think\>**: ...需要验证是否存在符合所有条件的车手...
**\<search\>**: ["F1最年轻积分车手纪录打破者", "F1三支不同车队席位调换", "德国站中游车队2008年领奖台", "耐力赛26号车手"]
**Search Results:** ...(correct answer is not included)

**\<think\>**: ...安东内利和维斯塔潘都涉及F1年轻纪录，但需要更具体信息。其中Arise Racing GT的26号车由Chaz Mostert驾驶，但需确认是否符合所有条件。
**\<search\>**: ["2023-2025年德国站中游车队领奖台", "26号赛车F1耐力赛", "F1三支不同车队席位调换", "Chaz Mostert车手履历"]
**Search Results:** ...

**\<think\>**: ...需要验证2023-2025年间是否有关于Chaz Mostert效力三支F1车队、席位调换、26号赛车使用的公开信息。可能存在未被检索到的2025赛季德国站中游车队领奖台记录，需补充检索2025赛季F1德国站官方成绩。
**\<search\>**: ["Chaz Mostert F1经历", "2023-2025德国站中游车队领奖台", "26号赛车F1车手", "F1三支不同车队席位调换"]
**Search Results:** ...[14]网页标题：...科维亚特在2022年纳斯卡杯系列赛中代表赫泽伯伯格车队出场三次，他的赛车使用与他在一级方程式赛车中相同的号码26...在赛季揭幕战赛百灵1000英里耐力赛（1000 Miles of Sebring）上，科维亚特获得第三名，首次登上WEC领奖台。...(correct answer is retrieved by "26号赛车F1车手")

**\<think\>**: ...丹尼尔·科维亚特符合耐力赛26号特征...
**\<answer\>**: ...丹尼尔·科维亚特✓... |

Blue: Exploring possibilities and widening scope. Orange: Narrowing down to incorrect candidates. Green: Narrowing down to right answer.
*Italics*: Faulty conflict resolution.

Table 10: DeepDiver leverages the SIS to compensate for gaps in flawed internal knowledge. Rather than prematurely providing an answer based on unverified or insufficient evidence, DeepDiver continues to intensify its search process, completing a total of 12 searches, progressively refining its findings until it identifies an acceptable answer.

| Problem | Throughout his career, he broke the record as the youngest driver to score points in Formula One history. He competed for three different top F1 teams and underwent two special seat swap arrangements. At the German Grand Prix, he helped a midfield team secure its first podium finish since 2008. After transitioning to endurance racing, he piloted the No.26 car and crafted new history across multiple events. Who is this driver? |
|---|---|
| Solution | Daniil Kvyat |
| R1-Prompting | **<think>**: ...Verstappen holds the youngest points record...Hülkenberg's age when scoring points in 2010 didn't break the record...Kvyat exclusion required (Scuderia Toro Rosso being midfield but didn't drive 26 in endurance)... **<search>**: ["F1 youngest points-scoring driver record", "Drivers with three F1 teams German GP podium", "2019 German Grand Prix podium teams", "WEC 26 car drivers F1 background"] **Search Results:** ...correct answer is retrieved by "2019 German Grand Prix podium teams" 

 **<think>**: ...Verstappen excluded...Kvyat...considered midfield....Alexander Albon...Need confirmation of other driver associations. **<search>**: ["Kvyat number of F1 teams served", "2019 German GP Toro Rosso midfield status", "F1 drivers WEC 26 car", "Daniil Kvyat endurance racing"] **Search Results:** ...(correct answer involved) 

 **<think>**: ...Kvyat fulfilling "first German GP podium by midfield team post-2008"...Other candidates like Hülkenberg Albon fail to meet all conditions. **<answer>**: ...Daniil Kvyat ✓... |
| Qwen7b-R1-Distill | **<think>**: ...Initial assessment suggests it may be Nico Hülkenberg,... **<search>**: ["Youngest point-scoring driver in F1 history", "Nico Hülkenberg seat swap history", "2014 German Grand Prix Sauber Nelson Piquet", "24 Hours of Le Mans Car 26 Hülkenberg"] **Search Results:** ...(correct answer is not included) 

 **<think>**: ...Nico Hülkenberg never achieved a podium finish, conflicting with the German GP third-place claim... **<search>**: ["2014 German Grand Prix Sauber team results", "Youngest point-scoring driver in F1 history", "24 Hours of Le Mans Car 26 driver Hülkenberg"] **Search Results:** ...(correct answer is not included) 

 **<think>**: *Contradiction resolution: F1 historical data shows the 2014 German GP's best midfield team result was Massa's third place (Ferrari), while Sauber's optimal result that race was Hülkenberg's third place. This likely stems from misinterpretation of "midfield teams" as "non-top teams", requiring explicit clarification in the response.* **<answer>**: ...Nico Hülkenberg ✗... |
| DeepDiver | **<think>**: ...Verify if there exists a driver meeting all criteria... **<search>**: ["Youngest F1 points record holder", "F1 seat changes across three teams", "Midfield team podium at 2008 German GP", "Endurance 26 car driver"] **Search Results:** ...(correct answer is not included) 

 **<think>**: ...Antonelli and Verstappen both relate to F1 youth records, but require more specific information... 26 is driven by Chaz Mostert... **<search>**: ["2023-2025 German GP midfield team podium", "No.26 car F1 endurance racing", "F1 seat changes across three teams", "Chaz Mostert racing career"] **Search Results:** ... 

 **<think>**: ...Verify...Chaz Mostert's participation with three F1 teams, seat changes, and No.26 car usage during 2023-2025. Potential unretrieved 2025 German GP midfield team podium record requires supplementary search of 2025 F1 German GP official results. **<search>**: ["Chaz Mostert F1 history", "2023-2025 German GP midfield team podium", "No.26 car F1 drivers", "F1 seat changes across three teams"] **Search Results:** ...[14] Webpage title:... Kvyat represented the Hezeburg team three times in the 2022 NASCAR Cup Series, and his car used the same number 26 as in Formula 1... In the season opener, the Sebring 1000-mile endurance race, Kvyat finished third and earned his first WEC podium....(correct answer is retrieved by "No.26 car F1 drivers") 

 **<think>**: ...Daniil Kvyat matches No.26 endurance racing characteristic... **<answer>**: ...Daniil Kvyat ✓... |

Blue: Exploring possibilities and widening scope. Orange: Narrowing down to incorrect candidates. Green: Narrowing down to right answer. *Italics*: Faulty conflict resolution.

Table 10: DeepDiver leverages the SIS to compensate for gaps in flawed internal knowledge. Rather than prematurely providing an answer based on unverified or insufficient evidence, DeepDiver continues to intensify its search process, completing a total of 12 searches, progressively refining its findings until it identifies an acceptable answer.

## Reasoning and Search Rounds of ProxyQA - DeepDiver

**Proxy Question:**
你能详细介绍一下世界上最辣的辣椒吗，包括它的起源、培育过程以及导致其极高辣度的因素？

**Response:**
1. 用户询问的是"世界最辣的辣椒"，首先需要明确具体是哪种辣椒，因为可能有很多种被称作"最辣"的，比如印度的Naga Viper、阿富汗的Bihun辣椒等，但需要确认权威来源。
2. 需要查找该辣椒的起源地，比如是否产自某个国家或地区，以及是否有特定的栽培历史。
3. 培育过程可能涉及抗病性、耐热性等改良，需要检索相关农业技术或育种论文。
4. 热度因素可能涉及辣椒素（如辣椒素、二氢辣椒素含量）、辣椒植株结构（如表皮厚薄）、生长环境（温度、湿度）等，需要具体数据支持。
5. 需要验证信息的权威性，比如是否被吉尼斯世界纪录或权威植物数据库认证。
6. 可能需要中英文搜索结合，覆盖更多资料来源。

...
**ROUND 1:** [世界最辣的辣椒具体品种, Naga Viper辣椒起源, 辣椒素含量与辣度关系, 辣椒育种技术最辣品种]

...
1. 根据最新检索结果，X辣椒被吉尼斯认证为269万SHU，而卡罗莱纳死神为220万，Naga Viper为135.9万，需明确当前世界最辣品种。
2. 需要补充X辣椒的培育细节，包括杂交品种及技术手段。
3. 需要验证不同辣椒的培育年限、抗病性等技术参数。
4. 热度因素部分需要具体分析辣椒素分布、植株结构等科学数据。
5. 需要对各辣椒的起源地进行交叉验证，例如Naga Viper是否确实源自英国。
6. 需要权威机构的最新数据，如2023年吉尼斯认证结果。
7. 用户可能对辣椒的化学成分与人类感官体验有更深入需求，需补充相关科学原理。

...
**ROUND 2:** [2023年最辣辣椒吉尼斯, X辣椒培育技术, 辣椒素分布与辣度关系, 辣椒育种抗病性, Naga Viper 起源地]

...
1. 根据最新检索结果，2023年吉尼斯认证的X辣椒辣度为269万SHU，超过卡罗莱纳死神的220万，成为新的世界之最，但需确认2024年是否有新纪录。
2. 网页[17]提到Naga Viper由英国培育，但2023年X辣椒认证后，需重新评估其地位。
3. 育种技术方面，需补充X辣椒的10年培育周期、杂交品种组合及抗病性改良。
4. 辣度因素中，网页[13][14][17]均提到辣椒素在胎座、隔膜等部位集中，但需要具体数据支撑。
5. 需要验证2024年是否有新的辣椒品种突破269万SHU。
6. 用户可能对辣椒的化学成分与神经科学的关联性感兴趣，可补充辣椒素对TRPV1通道的作用机制。

...
**ROUND 3:** [2024年最辣辣椒吉尼斯, X辣椒抗病性, X辣椒辣椒素胎座含量, TRPV1 辣椒素]
...

Figure 8: DeepDiver's reasoning process on ProxyQA involves expanding its search beyond a narrow scope to explore additional knowledge when faced with open-ended, challenging problems, improving its generalization through SIS-incentivized behavior.

## Reasoning and Search Rounds of ProxyQA - R1-Distill

**Proxy Question:**
你能详细介绍一下世界上最辣的辣椒吗，包括它的起源、培育过程以及导致其极高辣度的因素？

**Response:**
1. 用户需要了解世界上最辣的辣椒的详细信息，包括起源、培育过程和辣度因素。首先需要确定"世界最辣辣椒"的具体品种，常见候选包括卡罗来纳辣椒、印度魔鬼椒、龙舌兰辣椒等，但需要验证最新权威数据。
2. 起源部分需明确原产地（如美洲、亚洲）及驯化历史，需搜索"最辣辣椒原产地""辣椒驯化过程"等关键词。
3. 培育过程涉及选种、嫁接技术、抗病性改良等，需搜索"辣椒育种技术""高辣度辣椒栽培方法"等。
4. 辣度因素需区分自然基因（如CAPS含量）和环境因素（如温度、湿度），需搜索"辣椒辣度形成机制""环境对辣椒素影响"。
5. 需验证最新吉尼斯纪录数据，确认当前最辣品种是否为卡罗来纳死神（165万-325万斯科维尔）或某新型培育品种。
6. 需注意区分不同辣椒的辣度范围，避免将不同品种的数据混淆。

...
**ROUND 1:** [世界最辣辣椒品种吉尼斯纪录, 卡罗来纳死神辣椒起源培育, 辣椒素CAPS合成机制环境影响, 高辣度辣椒育种技术最新技术]
...

Figure 9: The R1-distilled model on ProxyQA mimics DeepSeek R1's behavior but shows overconfidence in its flawed internal knowledge, resulting in just one round of reasoning and searching.

## Answer Rounds of ProxyQA - R1-Distill

**Proxy Question:**

你能详细介绍一下世界上最辣的辣椒吗，包括它的起源、培育过程以及导致其极高辣度的因素？

---

**Response:**

目前公认的世界上最辣的辣椒是"辣椒X"（PepperX），其史高维尔辣度单位（SHU）检测值达到269万[1][2]。该品种由美国辣椒育种专家Ed Currie开发，以卡罗来纳死神辣椒为基础，通过杂交技术引入其他高辣品种特性[4]。

起源方面，辣椒原产于中美洲，现代栽培品种经过长期驯化。"辣椒X"的培育过程涉及多代杂交选育，Currie通过特殊处理（如硫酸铜种子消毒）和精准的田间管理，最终培育出这种扭曲皱缩的黄褐色果实[2][4]。

辣椒的辣度主要由辣椒素（capsaicin）及其衍生物含量决定，这些生物碱集中分布在白色"辣囊"组织中[2]。其合成受CAPS基因调控，该基因的表达在果实发育后期显著增强[5]。环境因素如温度（20-30°C最佳生长）和湿度也会影响辣椒素积累，Currie通过控制温室条件实现辣椒素最大化[4][7]。

培育技术上，该品种采用分阶段育苗：种子先在1%硫酸铜溶液中浸泡消毒后催芽，幼苗移植到营养土中定期施肥，成熟果实需在特定部位采收以保证辣度[4][7]。目前辣椒育种正向基因编辑方向发展，最新研究已发现调控辣椒素合成的关键转运蛋白基因[6]，为培育更高辣度品种提供新可能。

Figure 10: With limited reasoning and searching, the R1-distilled model produces an output that lacks informativeness, comprehensiveness, and depth.

**Answer Rounds of ProxyQA - DeepDiver**

Proxy Question:
你能详细介绍一下世界上最辣的辣椒吗，包括它的起源、培育过程以及导致其极高辣度的因素？

关于世界最辣辣椒（截至2024年）的详细介绍如下：

**1.**主要品种

**2024年吉尼斯认证最辣辣椒**

- **X**辣椒（**Pepper X**）：由美国辣椒培育专家埃德·柯里（Ed Currie）于2024年培育，经吉尼斯世界纪录认证为**269.3**万**SHU**（史高维尔辣度单位）。该辣椒被称作"辣椒界的核武器"，其黄绿色扁平扭曲的外观与"卡罗莱纳死神"（220万SHU）相比，辣度提升1.6倍[2][9][20]。
- 卡罗莱纳死神（**Carolina Reaper**）：该品种2012年获吉尼斯认证，平均辣度156.93万SHU，峰值220万，是首款突破200万SHU的商业辣椒[1][2][7]。
- **Naga Viper**：2011年在英国培育，辣度135.9万SHU，是首获吉尼斯认证的"世界最辣"辣椒[3][17]。

**2.**育种技术

- 长周期杂交：X辣椒育种历时10年，柯里每年进行100次杂交试验，通过将卡罗莱纳死神与"密歇根杂交辣椒"进行"精准嫁接"，使子房室辣椒素储存面积扩大[2][20]。
- 抗病性改良：育种过程中采用硫酸铜溶液浸种（1%浓度，10分钟）[7]，配合无土育苗技术（清水中浸泡5小时后，用25mL乙醇溶解0.5g辣椒粉提取辣椒素进行检测）[14]，有效防止土传病害。

**3.**辣度形成机制

- 辣椒素分布：辣椒素（辣椒碱）主要集中在辣椒的胎座、隔膜及果肉（占总辣椒素含量的70%以上）[23][24]。X辣椒通过基因改良，使胎座区域辣椒素浓度提升40%[24]。
- 表皮结构：其表面褶皱增加辣椒素接触面积，类似人类皮肤褶皱会增强疼痛感[13]。
- 辣椒素含量：1mg/kg的辣椒素约等于1600万SHU，X辣椒的辣椒素浓度达**5.771mg/g**（干重），远超普通辣椒[24][23]。

**4.**栽培与食用

- 种植技术：需保持土壤pH6.0-7.0，成熟期（90-120天）需每日**8**小时全日照，同时采用轮作制（避开茄科作物）[7][12]。
- 食用风险：2023年记录显示，食用30mg
  X辣椒可使人体血清素释放量激增，引发"辣椒素中毒"症状（胸闷、呼吸困难），甚至导致免疫系统超负荷[2][20][26]。

**5.**科学意义

- **TRPV1**通道激活：2021年诺贝尔奖得主大卫·朱利叶斯（David Julius）发现，辣椒素通过激活瞬时受体电位香草酸型**1**（**TRPV1**）通道，使神经末梢持续产生"灼烧感"[25][26]。X辣椒中辣椒素与二氢辣椒素（占比72%）的协同作用，使该通道被激活概率提升2.3倍[24]。

**6.**特殊用途

- 工业应用：X辣椒被用于高端辣椒酱，2023年上市24小时内售罄1000瓶，其辣椒素提取物还被开发为新型止痛剂候选[9][20]。
- 科研价值：其135.9万SHU的辣椒素提取物被用于癌症疼痛研究，2024年相关论文显示，高浓度辣椒素（>200万SHU）可使癌细胞线粒体应激反应增强48%[24][26]。

Figure 11: After several rounds of reasoning and searching, DeepDiver generates a result that covers various aspects of the given question, delivering extensive and comprehensive content with a wealth of knowledge.

## Iterative RAG Prompt

你是一名善于准确灵活地使用搜索工具、且富有解题策略和创造性的助手。我会给你一个问题，接下来通过多轮搜索工具调用解决问题，每一轮中你都需要先思考，然后再决定是否检索或者给出最终答案。如果你选择检索，我会返回给你检索结果然后再进行下一轮。

## 每一轮回复要求
每一轮你的回答需要包括两部分，第一部分是思考，第二部分是你的最终回答。

## 你的思考部分（<thinking>和</thinking>中间部分）：
1. 原则是尽可能的利用搜索引擎帮助解决问题或验证结论，每一轮要基于之前已有思维链条继续自我思考分析回答问题，要善于利用思维链条已有结论，不要重复推理和计算；
2. 如果需要检索，检索规划时，请避免基于内部知识进行草率、较大跳跃的推论，总是要假定还有其他可能，防止过早的缩小问题的范围；
3. 多来源交叉验证和冲突解决：强调对多个来源的交叉验证，尤其是当多个结果出现矛盾时，应该主动发起更多检索，比较不同来源的可信度，进行更深入的检索规划，而不是快速做决定、草率的选一个可能权威的来源；
4. 回溯和跳出局部假设：在遇到多次检索没有搜集到严格符合要求的信息的情况时不要轻易放弃，总是要考虑是否有其他可能性，能够重新评估假设，调整搜索策略，回溯和跳出局部假设，重新规划；
5. 基于一般搜索引擎的特点，要根据问题类型善于利用如拆解、细化以及相反的简化为更上位的概念、搜索原题原文、搜索类似问题等多个手段；
6. 你的思考使用语言尽量和用户问题语言保持一致，但你可以尝试多种语言的搜索语句，目前的搜索引擎主要以中文为主，所以英文问题也要尝试用一部分中文搜索语句。

## 你的最终回答部分（</thinking>后的部分）：
1. 如果需要进一步检索，则该部分为工具调用格式：web_search|{'search_queries': ['搜索语句1', '搜索语句2', ...]}，不要有任何其他多余的内容；
2. 如果不需要进一步搜索，则：
    - 综合整个思维链条给出信息量丰富、逻辑清晰的最后回复；
    - 除非用户要求，否则你最终回答的语言需要和用户提问的语言保持一致。

## 问答规范：
下面是一些回答问题的规范，如有必要可以在思考部分回顾与其可能相关规范（回顾规范时，要复述相应规范条例的对应片段，而不是指出基于第几条规范），以及在最终回复部分遵循这些规范：
    - 安全性：回答必须符合基本道德规范及主流价值观，体现人文关怀；不能复述用户问题中的敏感和不文明用词用语；谨慎给出不权威的影响用户重要决策的建议；
    - 完整性：用户问题的所有部分都应该在思考中考虑到；
    - 对于客观类的问答，如果问题的答案非常简短，可以适当补充一到两句相关信息，以丰富内容；
    - 对于长文生成类的问题（如写报告、论文、攻略），你需要解读并概括用户的题目要求，选择合适的格式，充分利用搜索结果并抽取重要信息，生成符合用户要求、极具思想深度、富有创造力与专业性的答案。你的写作章节和篇幅要尽可能延长，对于每一个要点的论述给出尽可能多角度的回答，务必信息量大、论述详尽。

## 输出格式：
思考部分用<thinking></thinking>标签完整包裹，</thinking>后接着输出最终回复或工具调用，不要有多余的不属于思考和最终回复两部分的前缀和后缀的解释和描述。输出示例如下：
<thinking>[你的思考过程...]</thinking>[你的最终回复或工具调用...]

[问题开始]
$query
[问题结束]

Figure 12: The prompt we designed to implement iterative RAG, which is used in WebPuzzle data tagging (Section 3.1), cold start data construction (Section 3.2), DeepSeek-R1 distillation, and multiple iterative RAG baselines (Section 4.1).

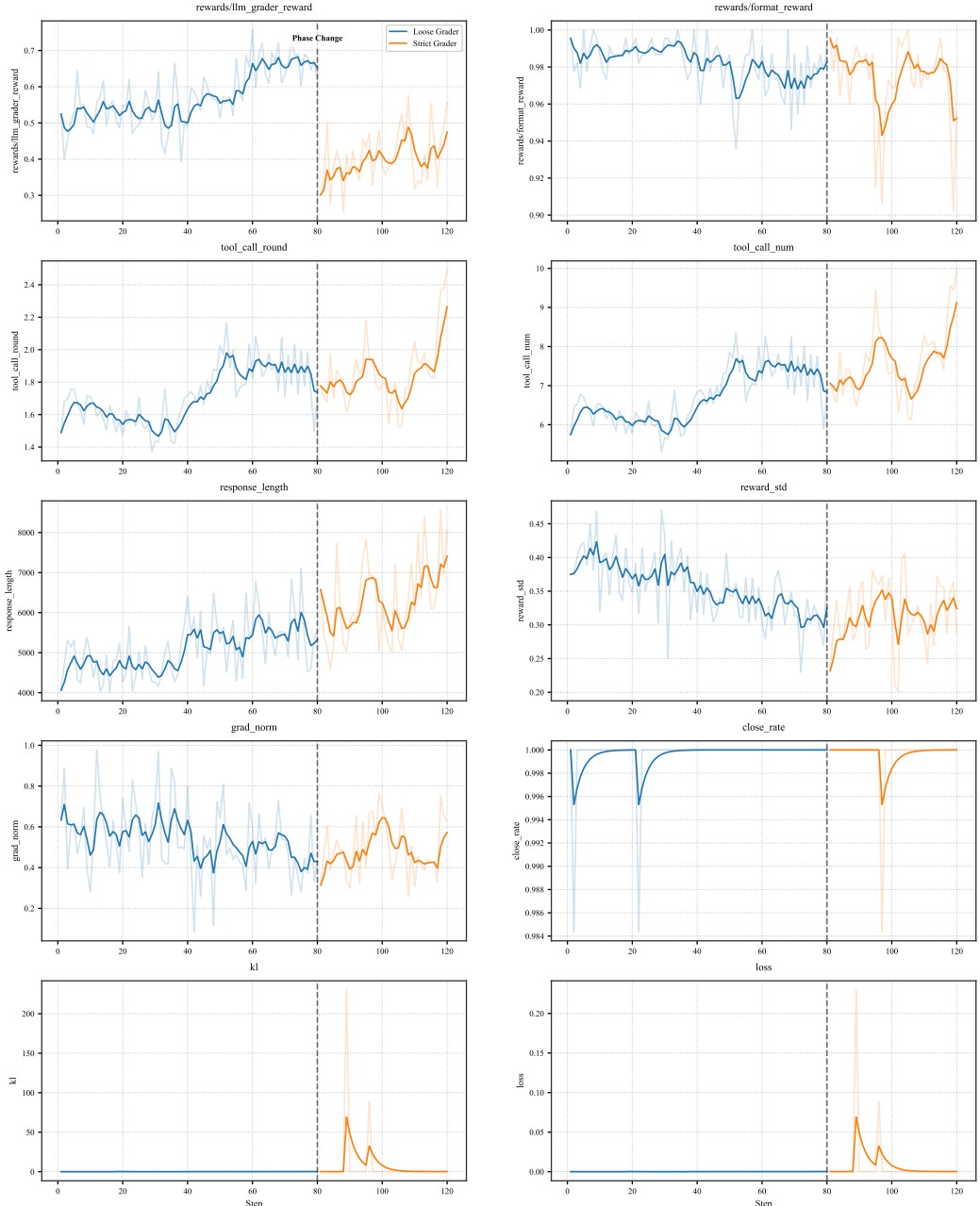

Figure 13: The detailed training curve of the DeepDiver-Qwen2.5-7B-Instruct model shows the trending of key metrics throughout the training process. These metrics include reward, tool call frequency, response length, reward standard deviation, gradient norm, KL divergence, and response completion rate.

## Loose Grader Prompt

你将看到一个用户问题和待评估的回答，我会给你参考答案和回答规范，你需要依据下面的要求对待评估回答进行分析和打分。

## 打分标准
1. 请以是否很好的回答用户问题为准则，参考答案只是辅助信息。
2. 待评估答案与参考答案的"关键语义含义"大致相同即可，不必严格遵照参考答案的文字表述。
3. 对于编号或分步骤的问题，完全可以接受步骤序号不同、划分粒度不同的情况，只要含义相同即可。
4. 对于列举类问题，如果待评估答案覆盖了参考答案中的一部分，另一部分有所遗漏，请根据正确覆盖的比例给出合理分数。
5. 对于数值类的答案在"得分标准"允许的误差范围内，都视为正确。
6. 参考答案的细节性要求有时过于严格，请不要严格遵照，如"必须..."“不能...”"一定要..."等，只要不影响给用户提供正确和完整的信息，可以忽略参考答案的严格要求。
7. 扣分情况：（1）参考答案的内容覆盖不完整，按遗漏比例酌情扣分（2）与参考答案有事实性冲突，应大幅度扣分。

基于参考答案和打分标准，按下面的等级对待评估回答给出1~10分。
10分：待评估回答的结论和参考答案完全一致，完全遵从回答规范。
8~9分：没有硬性问题，即待评估回答的结论和参考答案完全一致，基本遵从回答规范。
6~7分：正确性方面没有原则性问题，即待评估回答的结论与参考答案说法不冲突或互为包含关系，但可能对回答规范有一定漏考虑的地方，或推理思路可能有不清晰具体的问题。
5分：正确性有小问题，即待评估回答的结论和参考答案不完全一致，比如可能有漏考虑或多考虑的地方。
2~4分：正确性有不小的问题，即待评估回答的结论和参考答案有一定本质的区别，比如可能有理解错误、推理计算错误。
1分：待评估回答完全不可用，比如结论完全错误且与正确答案差距很大。

## 待评估内容
[用户问题开始]
$query
[用户问题结束]
[参考答案开始]
$solution
[参考答案结束]
[回答规范开始]
$checklist
[回答规范开始]
[待评估答案开始]
$response
[待评估答案结束]

## 你分两步输出你的结果：
第一步：分析。
基于上述给定各维度标准，以及打分评价方法，一步一步分析待评估回答属于评分标准中的哪一类，以及应该给出的分数。
该步骤输出格式为：第一步，分析：...
第二步：json输出。
为了方便解析，总结你第一步分析的结果，按如下json格式输出：
第二步，json输出：
{
"打分理由": "",
"得分": 1~10,
}

Figure 14: The prompt used for the loose grader, which assigns a score between 1 and 10 based on a single evaluation. A score of 6 or higher is considered correct. The design aims to enhance training signals and stabilize the learning process in the early training stage.

**Strict Grader Prompt**

**Strict Grader Prompt 1:**
你是一个资深的问答专家，根据用户问题、标准答案、答案checklist、问题回复这四项内容，评估问题回复的质量。
用户问题：$query
标准答案：$solution
答案checklist: $checklist
问题回复：$response

标准如下：
1. 根据标准答案，你判断问题回复的意思是否和标准答案一致，如果一致则正确，不一致则判为错误。注意：数值类的答案在答案checklist允许的误差范围内，都视为正确。
2. 根据答案checklist中描述的要求，衡量必须答对的部分与适当宽松的部分，你判断问题回复的答案是否符合要求，符合则正确，不符合则判为错误。
3. 只有同时符合前两条，问题回复才会最终被视为正确，否则问题回复最终被视为错误。

输出要求：
**第一步，思考**
以"第一步，思考："开始，鼓励你进行细致和审慎的推理和思考，直到你的思考过程已经完整详尽且逻辑严密，足够给出最终的评判结果，即可停止思考并给出评判结果。
**第二步，评判结果**
以"第二步，评判结果："开始，严格按照以下给定的字典格式输出最后的评估结果，不要输出带有"json"等话术，你的输出严格按照以下给定的字典格式，不要输出任何无关内容。
最终输出格式为：{"回复正确性": "正确"/"错误"}

---

**Strict Grader Prompt 2:**
假设你作为一位经验丰富的问答专家，你需要基于后续给定的用户问题、标准答案、答案checklist、问题回复，并结合两个核心维度，对回复内容的准确性进行评估。
用户问题：$query
...

评估标准如下：
1. 参照标准答案，你需评估问题回复是否与标准答案相吻合，若两者意思相同则视为正确，否则判定为错误。注意：如果在答案checklist中有提及允许的误差范围，数值类的问题回复只要在误差范围内，都视为正确。
2. 跟从答案checklist的要求，衡量问题回复，给出正确或错误的结论。
3. 前两点的答案如果都是正确，那么问题答案是正确的。否则，问题答案是错误的。
...

---

**Strict Grader Prompt 3:**
假设你是问答领域的资深专家，你将基于给定的用户问题、标准答案、答案checklist、问题回复，并结合以下提及的两个评测维度，对问题的回复进行综合评估。
用户问题：$query
...

接下来给出两个评测维度的标准，如下：
1. 你将标准答案作为参照，然后评估问题回复是否准确无误地反映了标准答案的内容，若相符则判断为正确，否则判断为错误。需要注意的是：计算类的结果如果有轻微误差，则视为正确。
2. 你需根据答案checklist对问题回复进行判断，若符合答案checklist的每一项要求则判定为正确，否则视为错误。
3. 只有前两条的结论都是正确，问题回复的最终评估结果才是正确，否则问题回复的最终结果是错误。
...

Figure 15: Different prompts used in the three rounds of validation for the strict grader. Each round of validation generating a binary classification of "correct" or "incorrect." The final result is determined by majority voting across the three evaluations. This grader is employed in the later stages of training to further enhance performance.

**Reflection & Correction Prompt:**
下面是一个大模型调用搜索工具进行多轮推理和检索回答问题的思维链条，我会给你该问题、思维链条和问题的标准答案。

[问题开始]
$query
[问题结束]
[思维链条开始]
$cot
[思维链条结束]
[问题答案开始]
$solution
[问题答案结束]

我需要你帮我评估并计数该思维链条中是否存在任何Reflection & Correction的模式，即模型发现了前面步骤中潜在的错误或遗漏，重新评估其已有思维链条，审视先前的假设，并明确地纠正或细化先前的推理步骤或结论。
请计数Reflection & Correction模式一共在思维链条中一共出现了多少次，如果不存在则计数为0。请先分析思维链条各部分是否符合该模式特征，并最后将该计数结果写在<count></count>中。

---

**Conflict Resolution Prompt:**
...
我需要你帮我评估并计数该思维链条中是否存在任何Conflict Resolution的模式，即在检索到的信息包含矛盾或不一致的情况下，模型能够发现矛盾并主动发起新的检索以解决冲突。
请计数Conflict Resolution模式一共在思维链条中一共出现了多少次，如果不存在则计数为0。请先分析思维链条各部分是否符合该模式特征，并最后将该计数结果写在<count></count>中。

---

**Verification & Denoising Prompt:**
...
我需要你帮我评估并计数该思维链条中是否存在任何Verification & Denoising的模式，即在给定有噪声或不相关的检索信息，模型分离出可信信息，从而执行去噪。
请计数Verification & Denoising模式一共在思维链条中一共出现了多少次，如果不存在则计数为0。请先分析思维链条各部分是否符合该模式特征，并最后将该计数结果写在<count></count>中。

Figure 16: Prompts used for automatically evaluating the occurrence counts of different behaviors in model outputs.

