# OpenReview forum: "DeepDiver: Adaptive Web-Search Intensity Scaling via Reinforcement Learning"
_NeurIPS.cc/2025/Conference — NeurIPS 2025 spotlight_

### Official Review · Reviewer_rFPC · 2025-07-03

**Clarity:** 3
**Significance:** 3
**Originality:** 3
**Rating:** 5
**Confidence:** 3

**Summary:**

The paper explores the task of open-web question answering to evaluate the ability of a model / agent on real-world information gathering and reasoning. The paper has two key contributions: 1) Webpuzzle which provides a more challenging real world benchmark (with both training and test sets) for real-world open-web question answering. 2) Deep-diver - an RL based framework which encourages higher search frequency and depth (by reward shaping i.e. rewarding GRPO rollouts which give correct answer while using search). Quantiative results show that Deep-diver performs favorably as compared to prior open-weight models.

**Questions:**

* In Sec. 3.2, the authors mention for cold start they use: "300 real-user questions from our deployed smart assistant". Can the authors please provide details on this?
* For iterative RAG mentioned in Sec. 3 for GRPO, what is the exact scaffold used for inference and training (e.g., basic ReACT loop, or more complex iterative multi-step scaffoldsOpenHands etc)
* The use of extra search call rewards is interesting, but did the authors explore other simpler alternatives such as just providing a small reward for solving with search etc.
* What is the maximum number of steps in each rollout during training and inference?
* For closed-source models in Table 1 only gpt4o is evaluated, can the authors also provide results with better models such as o3 or claude-4-sonnet. This will better help understand the performance on the proposed test set.
* Since it is a new benchmark, thorough evaluation across different baselines (including those with better scaffolds such as OpenHands) can help the reader better understand the difficulty of the proposed benchmark.
* Also can the authors provide more details on RL training, such as max context, max steps, the training algorithm, use of overlong filtering etc for GRPO.
* Finally, have the authors tried to compare performance of RL model trained on Webpuzzle with RL model trained on previous benchmarks and training datasets?

**Ethical Concerns:**

["NO or VERY MINOR ethics concerns only"]

**Final Justification:**

The authors have addressed most of my concerns. I will therefore raise the score.

**Limitations:**

yes

**Quality:**

3

**Strengths And Weaknesses:**

Strengths:
* The paper proposes a challenging Webpuzzle dataset for evaluating information seeking capabilities of agents in real-world scenarios.
*  The use of extra search call rewards in Sec. 3 is interesting.
* The paper has some nice insights with regards to correlation of agent behavior with final performance (such as increase in search frequency)
* The results in Sec. 5.2, for cross-lingual generalization are also interesting

Weaknesses:
* In Sec. 3.2, the authors mention for cold start they use: "300 real-user questions from our deployed smart assistant". Can the authors please provide details on this?
* For iterative RAG mentioned in Sec. 3 for GRPO, what is the exact scaffold used for inference and training (e.g., basic ReACT loop, or more complex iterative multi-step scaffoldsOpenHands etc)
* The use of extra search call rewards is interesting, but did the authors explore other simpler alternatives such as just providing a small reward for solving with search etc.
* What is the maximum number of steps in each rollout during training and inference?
* For closed-source models in Table 1 only gpt4o is evaluated, can the authors also provide results with better models such as o3 or claude-4-sonnet. This will better help understand the performance on the proposed test set.
* Since it is a new benchmark, thorough evaluation across different baselines (including those with better scaffolds such as OpenHands) can help the reader better understand the difficulty of the proposed benchmark.

---

> ### Author Rebuttal · Authors · 2025-07-31
>
> We appreciate your thoughtful review and interest in our work. Below, we clarify the concerns raised and will incorporate the suggested improvements in the final version.
>
> ---
>
> > In Sec. 3.2, the authors mention for cold start they use: "300 real-user questions from our deployed smart assistant". Can the authors please provide details on this?
>
> **Response:**
>
> The *"300 real-user questions from our deployed smart assistant"* refer to **open-ended** queries collected from internal team members interacting with our in-house assistant system, which uses DeepSeek-V3 and our internal search backend. Unlike WebPuzzle's close-ended QA format, these queries reflect diverse real-world information needs, such as:
>
> * *Should I buy an electric or gasoline car in 2025?*
>
> * *Any flagship TWS headphones worth recommending?*
>
> * *What’s your opinion on xxx event? What impact might it have?*
>
> These examples cover current events, product comparisons, and opinion-based analysis. We included them in training to **avoid overfitting to narrow, factoid-style QA tasks.**
>
> ---
>
> > For iterative RAG mentioned in Sec. 3 for GRPO, what is the exact scaffold used for inference and training (e.g., basic ReACT loop, or more complex iterative multi-step scaffolds like OpenHands)?
>
> **Response:**
>
> We apologize for the ambiguity in our original description.
>
> Our iterative RAG setup for GRPO uses a **basic ReAct-style loop**, implemented in-house. The inference pattern follows the standard *thought → search → thought →  … → answer* structure, closely aligned with the original ReAct formulation.
>
> For training, we combine this ReAct-style reasoning loop with the Hugging Face TRL library. GRPO and other RL algorithms are implemented via TRL, while the agent orchestration is handled by our own codebase.
>
> ---
>
> > The use of extra search call rewards is interesting, but did the authors explore other simpler alternatives such as just providing a small reward for solving with search etc.?
>
> **Response:**
>
> Thank you for the insightful question.
>
> Yes, we experimented with simpler reward schemes like that in the R1-searcher baseline, where a small bonus is given for solving with search. However, this led to **over-searching**—models learned to invoke the search tool even when unnecessary.
>
> Our design of the **extra search call reward** specifically addresses the **opposite problem** observed in early training: models were overly reliant on internal knowledge and hesitant to use external tools.
>
> Crucially, **correct answers with and without search receive equal terminal rewards**, avoiding long-term bias toward tool use. As shown in the table below, the frequency of this reward **naturally decays** during training, functioning as a **transient scaffold**:
>
> | Step Range |  Count | Percentage |
> | ---------- | ------------------ | ---------- |
> | 0–9        | 198                | 4.5%       |
> | 10–19      | 140                | 3.1%       |
> | 20–29      | 104                | 2.3%       |
> | 30–39      | 49                 | 1.1%       |
> | 40–49      | 42                 | 0.9%       |
> | 50–59      | 43                 | 1.0%       |
> | 60–69      | 27                 | 0.6%       |
> | 70–80      | 6                  | 0.1%       |
>
> This confirms the reward’s **temporary role** in overcoming early underuse of retrieval, without promoting long-term overdependence.
>
> ---
>
> > What is the maximum number of steps in each rollout during training and inference?
> >
>
> **Response:**
>
> We cap each rollout at a maximum of 7 reasoning steps, where each step follows a thought → search cycle in the ReAct-style loop.
>
> During each step, the model can issue multiple search queries in parallel, rather than being limited to one per step. This results in roughly 20–28 total queries per full rollout.
>
> ---
>
> > For closed-source models in Table 1 only gpt4o is evaluated, can the authors also provide results with better models ....
> >
> >Since it is a new benchmark, thorough evaluation across different baselines ....
> >
>
> **Response:**
>
> Thank you for the valuable suggestion. As recommended, we conducted additional evaluations on more advanced closed-source models, including **Claude-4-Sonnet**, **Gemini 2.5 Pro**, and **OpenAI’s o3 model**.
>
> We evaluated the models' *internal knowledge only* performance on our benchmark, with no retrieval support.
>
> **Accuracy by Category**
>
> | Category | o3 | claude-4-sonnet | claude-4-sonnet-thinking | gemini-2.5-pro |
> | --- | --- | --- | --- | --- |
> | **riddles** | **35.86%**  | 15.86%  | 24.14%  | 28.28%  |
> | **CrossQA** | 33.08% | 20.0%  | 20.77%  | **36.15%**  |
> | **Overall** | **34.55%**  | 17.82%  | 22.55%  | 32.00%  |
>
> **Accuracy by Difficulty**
>
> | Level | o3 | claude-4-sonnet | claude-4-sonnet-thinking | gemini-2.5-pro |
> | --- | --- | --- | --- | --- |
> | **medium** | **50.55%**  | 28.57%  | 39.56%  | **50.55%**  |
> | **hard** | **30.00%**  | 11.67%  | 15.0%  | 26.67 |
> | **Extreme Hard** | **25.21%**  | 11.76%  | 13.45% | 20.17%  |
> | **easy** | 20.00%  | **40.00%**  | 20.00% | 40%  |
>
> **Observation**: o3 and Gemini perform competitively with DeepSeek-R1 (32.7%), with o3 excelling on riddles and Gemini on CrossQA. Gemini, however, struggles more with harder tasks.
>
> We also tested Claude-4-Sonnet and o3 with our ReAct-style iterative RAG scaffold:
>
> **Accuracy by Category**
>
> | Category | o3 | claude-4-sonnet | claude-4-sonnet-thinking |
> | --- | --- | --- | --- |
> | **riddles** | 38.6% (1.05) | **41.4% (2.19)** | 40.7% (2.35) |
> | **CrossQA** | 53.8% (0.96) | 48.5% (1.48) | **58.5% (1.49)** |
> | **Overall** | 45.8% (1.01) | 44.7% (1.85) | **49.1% (1.95)** |
>
> **Accuracy by Difficulty**
>
> | Level | o3 | claude-4-sonnet | claude-4-sonnet-thinking |
> | --- | --- | --- | --- |
> | **medium** | **63.7% (0.62)** | 57.1% (1.66) | 62.6% (1.68) |
> | **hard** | 50.0% (1.0) | 50.0% (1.82) | **55.0% (1.98)** |
> | **Extreme Hard** | 29.4% (1.31) | 30.3% (2.05) | **35.3% (2.15)** |
> | **easy** | 60.0% (1.0) | **100.0 %(1.20)** | 60.0% (1.40) |
>
> *[Numbers in () indicate average search call rounds per example]*
>
> **Observation**: While Claude models lag behind o3 in the no-search setting, they gain 30+ points with retrieval, showing they are agent-compatible and effectively utilize external tools.
>
> **Evaluation of Off-the-Shelf Agent Products**
>
> We evaluated several commercial agent products and open frameworks on a **subset** of our benchmark (described in Appendix A.3: *Human vs. DeepDiver*), including:
>
> - **ChatGPT Deep Research Mode**
> - **Gemini Deep Research**
> - **OpenHands Cloud (open-source)**
>
> **Human vs. DeepDiver & Other Framework**
>
> | Model | **Accuracy** |
> | --- | --- |
> | **Human** | 44.0 |
> | **GPT-4o (Web Search Enabled)** | 60.0 |
> | **ChatGPT Deep Research** | **80.0** |
> | **Gemini Deep Research** | 52.0 |
> | **OpenHands Cloud** | 21.3 |
> | **DeepDiver** | 40.0 |
>
> **Insights:**
>
> ChatGPT Deep Research outperforms all others, including humans, but requires ~10 minutes per query, making it impractical for most real-time use cases. OpenHands Cloud underperforms mainly due to CAPTCHA issues blocking search/browse access.
>
> ---
>
> > Finally, have the authors tried to compare performance of RL model trained on Webpuzzle with ...
> >
>
> **Response:**
>
> Thank you for the thoughtful question.
>
> Yes, as presented in Section 5.2, we compared our DeepDiver with models trained on prior datasets:
>
> - **R1-Searcher**: trained on a Wikipedia-based search and data.
> - **DeepResearcher**: trained with open-Internet search but still based on Wikipedia corpora.
>
> Despite being trained entirely in Chinese, DeepDiver was evaluated on English benchmarks and search engines to test its cross-lingual generalization:
>
> | Model | WebPuzzle-en | BamBoogle | FRAMES | HotpotQA |
> | --- | --- | --- | --- | --- |
> | R1-Searcher | 13.7 (1.9) | 46.7 (2.0) | 25.3 (1.9) | 57.9 (2.3) |
> | DeepResearcher  | 15.0 (7.5) | 53.9 (7.1) | **33.6** (7.2) | 56.6 (4.4) |
> | DeepDiver-Qwen | **26.1** (14.7) | **56.8** (9.1) | 32.0 (14.2) | **58.4** (10.4) |
>
> **Key Observations:**
>
> - **DeepDiver** outperforms both prior methods on **WebPuzzle** by a great margin.
> - It also performs competitively on **wiki-based benchmarks**, despite **never being trained on English corpora**.
>
> We attribute this to **SIS**, which encourages more robust and tool-driven information gathering and verification.
> ### Behavioral Comparison: Information-Seeking Patterns
>
> As detailed in Appendix A.2, we compared the information-seeking behavior of each model using metrics from Section 2.2:
>
> | Methods | Model | WebPuzzle-en | BamBoogle | FRAMES | HotpotQA |
> | --- | --- | --- | --- | --- | --- |
> | **Reflection & Correction** | R1-Searcher | 0.04 | 0.03 | 0.04 | 0.02 |
> |  | DeepResearcher | 0.20 | 0.11 | 0.13 | 0.07 |
> |  | DeepDiver-Qwen | **0.45** | **0.25** | **0.28** | **0.27** |
> | **Conflict Resolution** | R1-Searcher | 0.06 | 0.02 | 0.02 | 0.02 |
> |  | DeepResearcher | 0.16 | 0.07 | 0.08 | 0.06 |
> |  | DeepDiver-Qwen | **0.31** | **0.23** | **0.32** | **0.18** |
> | **Verification & Denoising** | R1-Searcher | 0.17 | 0.18 | 0.13 | 0.10 |
> |  | DeepResearcher | 0.32 | 0.15 | 0.24 | 0.18 |
> |  | DeepDiver-Qwen | **1.72** | **1.80** | **1.60** | **1.54** |
>
> These findings reinforce two important points:
>
> 1. **WebPuzzle is more challenging** than traditional wiki-based datasets, requiring more complex reasoning and tool usage.
> 2. **DeepDiver** exhibits richer agent behaviors—iterative verification, reflection, and conflict handling—critical for open-domain performance.
>
> ---
>
> > Also can the authors provide more details on RL training, such as max context, max steps, the training algorithm, use of overlong filtering etc for GRPO.
> >
>
> **Response:**
>
> queries_per_step: 32
>
> group_size: 14
>
> temperature: 0.9
>
> kl_coef: 0.001
>
> max_tool_call_rounds: 7
>
> max_length: 22k
>
> overlong_reward: 0
>
> ---
>
> Thank you for your thoughtful question! We will revise our manuscripts according to your precious comments!

---

### Official Review · Reviewer_2jv9 · 2025-07-03

**Clarity:** 2
**Significance:** 4
**Originality:** 3
**Rating:** 5
**Confidence:** 4

**Summary:**

This paper investigates how to enable LLMs to perform adaptive, evidence-seeking behavior in open-domain question answering. To support this, the authors introduce WebPuzzle, a benchmark comprising both Wikipedia and open-web queries designed to elicit complex reasoning behaviors under noisy conditions in Chinese. They propose DeepDiver, a RL framework that trains LLMs to interleave reasoning and search, guided by a two-stage process involving supervised initialization and GRPO-based fine-tuning. A key emergent behavior, Search Intensity Scaling (SIS), allows models to dynamically adjust search depth based on task difficulty. Experiments show that DeepDiver-equipped 7B models outperform prompting and distillation baselines, approaching the performance of much larger models on multiple QA tasks.

**Questions:**

What’s the rationale behind not explicitly modeling the search?

In practice, how much impact would the search latency create with increasing search depth?

**Ethical Concerns:**

["NO or VERY MINOR ethics concerns only"]

**Final Justification:**

I've read authors' rebuttal, and continue to support the acceptance of this work.

**Limitations:**

yes in appendix

**Quality:**

3

**Strengths And Weaknesses:**

Strength
- The paper tackles an important and challenging problem of adaptive information seeking in real-world, noisy environments where static prompting and closed-domain SFT approaches fall short. The paper demonstrates a novel emergent behavior—Search Intensity Scaling (SIS)—where models increase search frequency and reasoning steps based on task difficulty, leading to substantial performance gains over prompting and distillation baselines.
- The authors introduce WebPuzzle, a new benchmark with 24k training and 275 expert-annotated test samples. It explicitly stresses noisy retrieval, ambiguity resolution, and reflective reasoning, offering a more realistic alternative to clean Wikipedia-style datasets.
- The DeepDiver framework is well-engineered, incorporating cold-start SFT, GRPO, and curriculum-aware reward shaping to stabilize and scale reinforcement learning on real-web tasks.
- The authors provide thorough empirical evidence, including accuracy on multiple Chinese QA benchmarks, information-seeking behavior counts, reward ablations, generalization English benchmarks etc.

Weakness
- The code, data, and models are not released, citing the internal review process in the checklist. This limits verification and reuse for now, and the release is not guaranteed.
- The test set size of 275 examples in WebPuzzle is relatively small for drawing strong statistical conclusions, especially given the complexity and variability of open-web search tasks.
- The reward function design is highly heuristic, i.e., manual engineering of multiple components, including staged loose-to-strict grading and auxiliary reward triggers.
- It is hard to understand the benefits of DeepDiver over existing agentic tool-augmented approaches, such as ReAct or WebGPT-style agents, which are not included as comparison systems in this work.

---

> ### Author Rebuttal · Authors · 2025-07-31
>
> Thank you for your thoughtful questions, we hope we could address these concerns with our below responses.
>
> ---
>
> > The code, data, and models are not released, citing the internal review process in the checklist. This limits verification and reuse for now, and the release is not guaranteed.
> >
>
> **Response:**
>
> We appreciate the reviewer’s concern. Both the code and data are **fully prepared for release**, and our internal review process has already been completed. We plan to make them publicly available **immediately following the paper’s acceptance announcement**.
>
> We would have been happy to release them via an anonymous repository during the rebuttal phase to facilitate transparency and reproducibility. However, **conference policy restricts releasing any links during the rebuttal periods**, and we aim to fully comply with these guidelines.
>
> We are fully committed to open-sourcing all artifacts to support verification and reuse.
>
> ---
>
> > The test set size of 275 examples in WebPuzzle is relatively small for drawing strong statistical conclusions, especially given the complexity and variability of open-web search tasks.
> >
>
> **Response:**
>
> We acknowledge the limited size of our benchmark. However, this is primarily due to the fact that all samples are **manually annotated and iteratively verified**, which is both time-consuming and labor-intensive. As shown in Section A.3, solving each problem requires an average of **9.28 webpages browsed per query**, and even under these conditions, human annotators only achieve **44.0% accuracy**.
>
> During the rebuttal period, we evaluated several advanced research agents on the 25 samples presented in Section A.3. The results are summarized below:
>
> | Model | **Accuracy** |
> | --- | --- |
> | **Human** | 44.0 |
> | **GPT-4o (Web Search Enabled)** | 60.0 |
> | **ChatGPT Deep Research** | **80.0** |
> | **Gemini Deep Research** | 52.0 |
> | **OpenHands Cloud** | 21.3 |
> | **DeepDiver** | 40.0 |
>
> As the table shows, **ChatGPT Deep Research** significantly outperforms both humans and other automated systems on this subset. However, solving each query typically takes **over 10 minutes**, even for state-of-the-art models. This highlights the difficulty of the task and the cost of high-quality annotation—making a benchmark size of **275 samples** a practical and acceptable compromise.
>
> Moreover, we note that several widely-used benchmarks—such as **WSC273**, **CB (SuperGLUE)**, **HumanEval**, and various subtasks in **BBH**—also contain fewer than 275 examples, reinforcing the precedent for using datasets of this scale in challenging reasoning tasks.
>
> That said, we agree with the reviewer on the importance of statistical robustness. In the revised version, we will include **confidence intervals** (CI) for key metrics to better quantify variability. We also plan to incorporate **pass@k** and **avg@k** metrics with a number of evaluation trials significantly larger than *k*, to provide a more comprehensive view of model performance.
>
> Thank you again for this valuable suggestion.
>
> ---
>
> > The reward function design is highly heuristic, i.e., manual engineering of multiple components, including staged loose-to-strict grading and auxiliary reward triggers.
> >
>
> **Response:**
>
> Thank you very much for this excellent question. Reward design for reinforcement learning in open and complex information-seeking tasks remains a relatively new challenge. Our reward design draws not only on experiences from previous works but, more importantly, on adjustments made in response to "reward hacking" phenomena observed in our experiments.
>
> 1. **Loose-to-Strict Reward Scheduling**
>
>     Initially, we employed a strict reward rubric based on LLM-as-Judge assessments. However, we quickly discovered that such a reward scheme led to a high prevalence of zero-reward episodes, preventing effective gradient propagation. Specifically, when the model’s early outputs were consistently poor, the computed advantage:
>
>
> $$
> \hat{A}_{i,t} = \frac{r_i - \mathrm{mean}(r)}{\mathrm{std}(r)},
> $$
>
> collapsed due to all-zero rewards triggers the zero adavantage. We also attempted curriculum learning by filtering for easier examples in early training, but this did not overcome the zero-reward issue.
>
> To address this, we designed a **loose reward** function that relaxed evaluation criteria, allowing the model to receive credit for partially correct or approximately matched outputs. This approach helped bootstrap the model’s learning. Once stable training progress was achieved, we transitioned to the **strict** reward rubric to further refine and push performance beyond this initial plateau.
>
> 1. **Auxiliary Tool-Use Reward**
>
>     Our motivation for introducing **extra** **search call rewards** is not to encourage excessive use of the search tool, but rather to **de-bias the model’s early reluctance to use external tools**. In the initial phase of training, we observed that the model **tends to over-rely on its internal knowledge and avoids invoking external tools**, even when they would be helpful. The additional reward is designed to encourage exploration of retrieval actions during this early phase.
>
>     Importantly, in contrast to R1-searcher, **our reward design does not give higher returns to search-based solutions when the internal knowledge is sufficient**. Instead, **we treat search-free and search-based correct trajectories equally**, which helps the model avoid unnecessary search calls once it becomes more capable.
>
>     To better understand the effect of our reward scheme, we analyzed the frequency of the extra search-call reward (value = 3.0) throughout training. Every 10 training steps (each step producing 448 trajectories), we measured how often this reward was triggered. As shown below, the reward is naturally **phased out over time**, confirming its role as a **transient scaffold** in early learning:
>
>     | Step Range | Reward = 3.0 (count) | Percentage |
>     | --- | --- | --- |
>     | 0–9 | 198 | 4.5% |
>     | 10–19 | 140 | 3.1% |
>     | 20–29 | 104 | 2.3% |
>     | 30–39 | 49 | 1.1% |
>     | 40–49 | 42 | 0.9% |
>     | 50–59 | 43 | 1.0% |
>     | 60–69 | 27 | 0.6% |
>     | 70–80 | 6 | 0.1% |
>
>     This trend shows that the extra search reward is **adaptive**, mainly active in the early training stages to de-bias the initial underuse of retrieval, and naturally diminishing as the model matures.
>
>     We will clarify this behavior and the motivation in the revised version of the paper.
>
>
> ---
>
> > It is hard to understand the benefits of DeepDiver over existing agentic tool-augmented approaches, such as ReAct or WebGPT-style agents, which are not included as comparison systems in this work.
> >
>
> **Response:**
>
> Our framework is indeed aligned with the **ReAct paradigm**. Each query is handled through a single trajectory that alternates between *reasoning* and *acting*—i.e., thinking, issuing a search query, processing the retrieved results, and planning the next step—following the core principles of ReAct.
>
> In fact, all tool-augmented baselines compared in our paper are **built on top of the ReAct framework**, including systems that perform multi-step reasoning and tool invocation. Our contribution lies in extending this framework with improved **search control**, **tool-use precision**, and **learning signals from reward models**, enabling better performance on complex queries that require deep research.
>
> While WebGPT-style agents focus on browsing based instead of searching / RAG agent, therefore not included in our baselines.
>
> ---
>
> > What’s the rationale behind not explicitly modeling the search?
> >
>
> Thank you for the insightful question. We distinguish between two types of tools in agent environments: those that can be effectively simulated by LLMs, and those that cannot.
>
> Tools like software applications or APIs often involve **deterministic interaction patterns** and do not require **real-time factual grounding**. These can be simulated by LLMs to teach agents how to operate them.
>
> Search engines, however, are fundamentally different. While the interaction logic—such as issuing queries or selecting snippets—can be simulated, the retrieved content cannot. Search results depend on real-time, external, and verifiable information that lies beyond the model’s parameters. Simulating such content would result in fabricated or outdated information, defeating the purpose of using a search engine.
>
> Crucially, training robust agents requires more than just learning to issue queries—it requires iteratively **verifying information, resolving conflicts across sources, and reasoning over real-world content**. These abilities depend on exposure to authentic, up-to-date information, which cannot be reliably simulated.
>
> In summary, although the interface behavior of search tools can be mimicked, the content’s factuality and variability make real search access essential for training agents with strong reasoning and verification capabilities.
>
> ---
>
> > In practice, how much impact would the search latency create with increasing search depth?
> >
>
> **Response:**
>
> Search latency primarily depends on **concurrency** during execution. For example, with fewer than 200 concurrent requests, the Bocha search API typically returns results within **9 seconds**. However, at higher levels of concurrency, latency may increase due to **queuing or retry mechanisms** on the API side.
>
> As search depth increases, **inference latency** can also rise, particularly in multi-turn ReAct-style trajectories. To mitigate this, we leverage **prefix caching** during decoding, which is especially effective in scenarios involving repetitive context patterns. In our tests, prefix caching improved inference **throughput by approximately 50%**, helping reduce latency as reasoning chains grow longer.

---

### Official Review · Reviewer_ucvZ · 2025-07-03

**Clarity:** 3
**Significance:** 3
**Originality:** 3
**Rating:** 5
**Confidence:** 3

**Summary:**

The paper studies the task of question answering via web information retrieval using Large Language Models. The authors provide a new benchmark dataset, WebPuzzle, testing the LLMs ability to navigate and reason about noisy information from websites. Using this, they train DeepDiver, a 7B LLM that reaches similar performance to much larger models on such task. For this, they propose a tailored RL pipeline to incentivize search intensity and generalize to new domains. Moreover, they analyze various evaluation aspects such as the relationship between search intensity and problem difficulty.

**Questions:**

- From my understanding, only 2k samples (demonstrations?) were used for SFT, while 5k samples (prompts?) for RL. Did the authors ablate the chosen split? In particular, I wonder how would the performance vary as a function of the amount of SFT data. E.g. what would have happened with a single SFT phase on 7k samples.
- Related to the above, online RL (via GRPO) seems crucial for performance but also the main computational bottleneck to scale the approach to more samples. Did the authors consider offline baselines (such as DPO) for sample efficiency?

**Ethical Concerns:**

["NO or VERY MINOR ethics concerns only"]

**Final Justification:**

The paper presents a novel relevant benchmark dataset and a performant LLM agent for the Web search task. The overall training pipeline is described, making it a nice contribution to the community.

**Limitations:**

yes

**Paper Formatting Concerns:**

I have not noticed any formatting issue.

**Quality:**

3

**Strengths And Weaknesses:**

Strenghts:
- The paper studies a relevant and significant problem, the contributions are clear and the overall exposition is good.
- The WebPuzzle dataset looks like a relevant benchmark that looks complex enough to separate and differentiate different LLM agents and approaches
- DeepDiver looks like a strong LLM agent for WebSearch
- The explanation and ablations of the RL method used for training DeepDiver is a valuable contribution to the community
- Experiments' analyses are somewhat extensive and informative.

Weaknesses:
- I think the authors could have given more details on the used RL algorithm and training procedure, with training metrics and ablations on the choices made.

---

> ### Author Rebuttal · Authors · 2025-07-31
>
> We sincerely thank the reviewer for the thoughtful and constructive feedback. Below, we address each of the points raised in detail.
>
> ---
>
> > I think the authors could have given more details on the used RL algorithm and training procedure, with training metrics and ablations on the choices made.
> >
>
> **Response:**
>
> Thank you for pointing this out. We have provided the key hyperparameters used during RL training in Appendix E.8, now summarized below for clarity:
>
> | Parameter | Value |
> | --- | --- |
> | Rollouts per Sample | 14 |
> | Sampling Temperature | 0.9 |
> | Batch Size | 32 |
> | Learning Rate | 1e-6 |
> | Number of Epochs | 1 |
> | KL Divergence Coefficient | 0.001 |
> | Max Tool Call Rounds | 7 |
> | Training Hardware | 1 machine with 8 × H20 GPUs |
> | Chinese Search Engine | Bocha |
> | English Search Engine | LangSearch |
> | Search Results Retained | Top 2 per query |
>
> Our choice of hyperparameters was informed by prior research, practical heuristics from relevant blogs, and official TRL guidelines and recipes. We evaluated multiple configurations and selected the ones that aligned best with trends in existing literature. For example, based on the existing experience of the community, the learning rate around 1e-6 [1,2], and the trend for the kl coefficient is to decrease gradually or even be omitted [3,4].
>
> ---
>
> > From my understanding, only 2k samples (demonstrations?) were used for SFT, while 5k samples (prompts?) for RL. Did the authors ablate the chosen split? In particular, I wonder how would the performance vary as a function of the amount of SFT data. E.g. what would have happened with a single SFT phase on 7k samples.
> >
>
> **Response:**
>
> We did in fact evaluate a baseline trained with the full 7k r1 teacher dataset using SFT, which we denote as **R1-Distill** in Table 1. As detailed in Appendix E.5, this model uses the entire 2k + 5k sample set in a single supervised fine-tuning stage.
>
> Compared to prior works that often compare their works with prompting-based methods, or RFT self-training baselines after the cold-start checkpoints, our R1-Distill model represents a stronger baseline by leveraging full teacher supervision. While it performs well, it still falls short of the RL-trained model—particularly on out-of-distribution tasks such as **Bamboogle**, as discussed in Section 4.4.
>
> Regarding the use of 2k samples for cold start and whether we experimented with other split ratios, we did not conduct such experiments. This is because the experimental cost of RL training is relatively high. The choice of cold-start data size was primarily based on experiences from previous literature [5], which suggested a data volume in the thousands, and then made our decision based on our overall data volume.
>
> ---
>
> > Related to the above, online RL (via GRPO) seems crucial for performance but also the main computational bottleneck to scale the approach to more samples. Did the authors consider offline baselines (such as DPO) for sample efficiency?
> >
>
> **Response:**
>
> We agree that offline methods like DPO offer a promising direction for improving sample efficiency. However, applying DPO to our multi-turn tool-use task poses specific challenges. Effective supervision requires carefully curated positive–negative pairs that capture not only correctness of the final answer, but also proper reasoning, tool invocation, and formatting.
>
> To explore this, we created 5k r1 examples consisting of correct–incorrect output pairs and used them to train **Qwen2.5-7B-Instruct** via DPO. However, the resulting model often failed to follow the required formatting for reasoning and tool usage.
>
> We also applied DPO starting from the cold-started Qwen2.5-7B-instruct checkpoint, but did not observe improvements over the base model. We hypothesize two main reasons for this:
>
> 1. The DPO training signal, which is applied at the sequence level, may be too coarse for our long (22k-token) and complex tasks that require step-by-step tool usage.
> 2. Due to time constraints of the rebuttal period, we did not have the opportunity to explore alternative DPO configurations or tune the relevant hyperparameters, and took more fine-grained DPO algorithm implementation. For example when the LLM is sampled multiple times with various correct and incorrect samples, we didn't take into care consideration that choose the most suitable positive-negative pairs (using tricks such as ranking, classification etc. or using synthesised negative samples for more a  contrastive learning objective), but simply randomly select one correct and one negative sample for the training.
>
> We acknowledge the potential of DPO and plan to explore it more thoroughly in the revision version. We greatly appreciate the reviewer’s insightful suggestion.
>
> ---
>
> **Reference**
>
> [1] Search-r1: Training llms to reason and leverage search engines with reinforcement learning.
>
> [2] R1-searcher: Incentivizing the search capability in llms via reinforcement learning.
>
> [3] DAPO: An open-source llm reinforcement learning system at scale.
>
> [4] Simplerl-zoo: Investigating and taming zero reinforcement learning for open base models in the wild.
>
> [5] Deepseek-r1: Incentivizing reasoning capability in llms via reinforcement learning.

---

### Official Review · Reviewer_5N3z · 2025-07-05

**Clarity:** 3
**Significance:** 3
**Originality:** 3
**Rating:** 4
**Confidence:** 4

**Summary:**

The paper investigates LLMs capability for open-web retrieval and proposes letting the model learn to perform proactive, multi-step searches when solving open-domain questions. The authors release WebPuzzle (24k training examples; 275 human-curated test questions), a dataset explicitly designed to measure an LLM’s ability to retrieve and reason over information in a realistic, noisy and fragmented web environment. They further introduce the DeepDiver framework: after a SFT warm-start, the system is refined with reinforcement learning (GRPO) to cultivate a policy that decides when and how much to search. A key claimed feature is the emergent SIS ability. Experiments with 7B parameter base models show that DeepDiver nearly matches R1 on real-time web QA benchmarks, while raising the average number of retrieval rounds from ~1.7 to > 2.5. The paper also analyses the training curriculum and the strategy’s generalisation to tasks such as open-ended long-form generation.

**Questions:**

1. Most of the training samples in the WebPuzzle dataset are synthesized through LLM. How do the authors ensure that these synthetic problems realistically reflect the complexity of real user search tasks?
2. DeepDiver’s RL training uses complex reward design and training scheduling. Can the authors share the sensitivity of key training settings?

**Ethical Concerns:**

["NO or VERY MINOR ethics concerns only"]

**Quality:**

3

**Strengths And Weaknesses:**

Pros:

1. The paper constructs a new WebPuzzle dataset to evaluate knowledge retrieval and reasoning in an open network environment.

2. The overall scores of Qwen2.5-7B and Pangu-7B trained by DeepDiver are almost equal to or even partially surpass those of the DeepSeek-R1 model, which has two orders of magnitude larger parameters.

Cons:
1. Although the paper claims that SIS is a major innovation, this capability is essentially a model behavior promoted by reward design, and is conceptually an extension of existing ideas. Using reinforcement learning to optimize LLM retrieval decisions is not new. In recent years, there have been works that use RL to let the model learn when to retrieve and how many steps to retrieve. The main contribution of the author's method is that it is applied to more complex environments and the number of model retrieval times is observed to increase. This is more of a phenomenon report rather than a new algorithmic principle.

2. The paper repeatedly describes SIS as a feature that emerges automatically after RL training, but in fact the authors explicitly add rewards to the model to encourage multiple searches. Calling this behavior "emergent" under the influence of rewards is a bit exaggerated and may mislead readers, as if the model has spontaneously evolved this skill.

---

> ### Author Rebuttal · Authors · 2025-07-31
>
> **Response to Reviewer 5N3z**
>
> ---
>
> Thank you very much for your thoughtful and constructive feedback. We truly appreciate your careful reading of our work and your insightful comments regarding the novelty of SIS and the reward design. Please allow us to clarify and expand on the points you raised.
>
> > Although the paper claims that SIS is a major innovation, this capability is essentially a model behavior promoted by reward design, and is conceptually an extension of existing ideas. Using reinforcement learning to optimize LLM retrieval decisions is not new. In recent years, there have been works that use RL to let the model learn when to retrieve and how many steps to retrieve. The main contribution of the author's method is that it is applied to more complex environments and the number of model retrieval times is observed to increase. This is more of a phenomenon report rather than a new algorithmic principle.
> >
> > The paper repeatedly describes SIS as a feature that emerges automatically after RL training, but in fact the authors explicitly add rewards to the model to encourage multiple searches. Calling this behavior "emergent" under the influence of rewards is a bit exaggerated and may mislead readers, as if the model has spontaneously evolved this skill.
> >
>
> **Response:**
>
> We apologize if our explanation was unclear and led to the impression that our approach is merely a restatement of prior work or that the behavior we observe is entirely driven by reward shaping.
>
> While it is true that our method includes an auxiliary reward to encourage search actions, our primary objective is to **mitigate the model’s initial reluctance to use external tools**, which we observed to be a consistent bias during early training. Specifically, in the early phases, the model tends to over-rely on internal knowledge—even when external retrieval is necessary (for example the questions about latest news). The auxiliary reward is thus introduced to encourage exploration of retrieval behaviors during this critical learning phase.
>
> Formally, the condition for extra rewards is defined as:
>
> $$
> \forall i \in G, \mathcal{S}_i = 0 \implies \mathcal{C}_i = 0, \quad \exists j \in G \text{ such that } \mathcal{S}_j = 1 \text{ and } \mathcal{C}_j = 1,
> $$
>
> where G is the group of rollouts, S_i indicates whether the i-th rollout uses a search engine, and C_i indicates success.
>
> We apply the extra reward with
>
> $$
> \mathcal{E}_i =
> \begin{cases}
> 1.0 & \text{if } \mathcal{S}_i = 1 \text{ and } \mathcal{C}_i = 1, \\
> 0.0 & \text{otherwise}.
> \end{cases}
> $$
>
> This means that when no **search-free** rollouts solve a problem but **at least** one search-enabled rollout succeeds, we assign an additional reward of 1.0 to the successful search-enabled solutions.
>
> This design **differs from prior work** such as R1-Searcher, which provides persistent additional rewards (+0.5) for using external tools. In contrast, **our reward scheme does not prefer search-based over search-free solutions when both yield correct results**. This ensures that the model does not develop a dependency on tool use, but instead learns to call tools only when beneficial.
>
> To better understand the effect of our reward scheme, we analysed the frequency of the special search-call reward (value = 3.0) throughout training. Every 10 training steps (each step producing 448 trajectories), we measured how often this reward was triggered. As shown below, the reward is naturally **phased out over time**, confirming its role as a **transient scaffold** in early learning:
>
> | Step Range | count | Percentage |
> | --- | --- | --- |
> | 0–9 | 198 | 4.5% |
> | 10–19 | 140 | 3.1% |
> | 20–29 | 104 | 2.3% |
> | 30–39 | 49 | 1.1% |
> | 40–49 | 42 | 0.9% |
> | 50–59 | 43 | 1.0% |
> | 60–69 | 27 | 0.6% |
> | 70-80 | 6 | 0.1% |
>
> From this, we highlight two key observations:
>
> 1. Even in the earliest stages, only a small fraction of trajectories receive the additional reward.
> 2. The reward is **naturally phased out** over time and becomes essentially inactive after approximately 30 steps.
>
> Despite this, as shown in **Figure 4** of our manuscript, the number of tool-use rounds continues to grow significantly, with a particularly sharp increase observed during steps 80–120—**well after the auxiliary reward is no longer in effect**. This pattern supports our view that **SIS** is **not merely a result of explicit reward shaping**, but rather an **emergent behavior** developed by the model during RL training. The model learns to leverage external tools to compensate for the limitations of its internal knowledge—**a behavior that persists and strengthens even in the absence of direct incentives**.
>
> We hope this clarifies the motivation and effect of our reward design, and why we consider SIS to be an emergent and meaningful outcome of our method.
>
> ---
>
> > Most of the training samples in the WebPuzzle dataset are synthesized through LLM. How do the authors ensure that these synthetic problems realistically reflect the complexity of real user search tasks?
> >
>
> Thank you for this important and insightful question. We fully agree that ensuring the realism and relevance of synthetic data is critical, particularly when modeling open-domain search and reasoning tasks. We address this concern through three complementary perspectives:
>
> 1. **From the Perspective of Data Curation and Generation**
>
>     WebPuzzle are not generated from templated rules or knowledge bases, but rather through **LLMs grounded in open-web content**.
>
>     - Cross-page QA: LLMs extract real web pages and then synthesize inverted, multi-hop questions that require reasoning across multiple sources. This goes beyond conventional Wikipedia-based QA datasets and aims to reflect the structure and ambiguity of real-world web search tasks.
>     - Riddle-style Question: LLMs are prompted to create riddles or obfuscated queries based on unique attributes of entities. This introduces indirect phrasing and semantic ambiguity, simulating the kinds of vague queries users often make online.
>
> 2. **From the Perspective of Human Evaluation**
>
>     To assess and improve realism, we conducted **manual filtering** during test set construction. From an initial pool of 500 synthetic examples, human annotators reviewed each item to:
>
>     - Eliminate samples that **cannot be solved within the Chinese internet environment**.
>     - Remove any questions that **do not adequately reflect the complexity** or ambiguity of real user queries.
>
>     This resulted in a curated test set of 275 examples. Notably, most exclusions were due to regional search constraints, while only a small portion were rejected for lacking sufficient complexity. This suggests that the majority of LLM-generated queries do capture key aspects of real-world difficulty.
>
> 3. **From the Perspective of Model Generalization**
>
>     We also validate the quality of WebPuzzle indirectly through **cross-dataset generalization**. Specifically, after training on WebPuzzle, our DeepDiver demonstrates strong performance on **Bamboogle**—a manually crafted dataset designed for evaluating real-world internet search capability. We believe this strong transfer supports the conclusion that WebPuzzle enhances model reasoning in ways that generalize beyond synthetic tasks.
>
> ---
>
> > DeepDiver’s RL training uses complex reward design and training scheduling. Can the authors share the sensitivity of key training settings?
> >
>
> **Response:**
>
> We are happy to share the insights we have gathered through extensive experimentation, including many unsuccessful trials.
>
> 1. **Loose-to-Strict Reward Scheduling**
>
>     Initially, we employed a strict reward rubric based on LLM-as-Judge assessments. However, we quickly discovered that such a reward scheme led to a high prevalence of zero-reward episodes, preventing effective gradient propagation. Specifically, when the model’s early outputs were consistently poor, the computed advantage:
>
>
> $$
> \hat{A}_{i,t} = \frac{r_i - \mathrm{mean}(r)}{\mathrm{std}(r)},
> $$
>
> collapsed due to all-zero rewards triggers the zero adavantage. We also attempted curriculum learning by filtering for easier examples in early training, but this did not overcome the zero-reward issue.
>
> To address this, we designed a **loose reward** function that relaxed evaluation criteria, allowing the model to receive credit for partially correct or approximately matched outputs. This approach helped bootstrap the model’s learning. Once stable training progress was achieved, we transitioned to the **strict** reward rubric to further refine and push performance beyond this initial plateau.
>
> 1. **Auxiliary Tool-Use Reward**
>
>     As discussed above, we found that LLMs exhibit a strong early bias toward using internal knowledge. Our auxiliary reward is designed solely to counteract this reluctance during early training. It is deactivated naturally during the middle training stage.
>
> 2. **Cold-Start Data Selection**
>     - We experimented with prompt prefilling strategies that concat phrases such as *“Wait, I need to search again to verify my conclusion”*. While this led to slight gains after SFT (around +0.5–1.5 points), the improvements did not persist after RL. Post-hoc analysis confirmed that such “s1-like” patterns were rarely used after RL, suggesting that forced prompting **encourages the search** does not translate into sustainable behaviors and performance gain.
>     - We also explored cold-start fine-tuning on the WebPuzzle dataset alone. However, without mixing in general reasoning datasets, the model’s performance was significantly worse. We found that exposure to diverse, long-context reasoning tasks during cold start  helped bootstrap the model’s ability to reason and reflect.
>
>     We hope these insights provide a clearer view into our training strategy and its sensitivity. We thank the reviewer again for encouraging us to share these details.

---

> > ### Author Response · Authors · 2025-08-09
> >
> > Dear reviewer 5N3z,
> >
> > Thank you for the thoughtful and constructive review. We provide detailed, point-by-point responses and additional clarifications addressing each concern. We welcome further discussion and, if our responses resolve the issues, would appreciate your reconsideration of the assessment. We are committed to incorporating your suggestions to strengthen the manuscript.
> >
> > Best regards,
> >
> > Submission 14894 Authors

---

### Note · Authors · 2025-08-15

We sincerely thank the AC and all reviewers for their thoughtful feedback and constructive input during the rebuttal period. We appreciate the opportunity to improve our work. Below, we summarize the major improvements made and key reviewer concerns addressed:

---

#### 1. **Clarifying "Emergent" Search Intensity Scaling (SIS)** (Reviewer 5N3z)

- **Reward Design**: Clarified that the auxiliary search reward is a temporary scaffold to counter initial tool reluctance, not a persistent bias. Training data show it phases out (0.1% of trajectories after step 70) while SIS persists and strengthens.
- **Evidence**: Even after reward vanishes, search rounds rise sharply in later stages (steps 80–120), confirming SIS emerges from RL, not reward shaping.

#### 2. **RL Training Details & Sensitivity** (Reviewers ucvZ, rFPC)

- **Hyperparameters**: Shared GRPO settings and empirical notes.
- **Reward scheduling**: Explained early zero-reward collapse and how relaxed initial criteria bootstrap training before refining with strict evaluation.

#### 3. **Benchmark** (Reviewers rFPC, 2jv9)

- **Closed-Source Models**: Added results for o3, Claude-4-Sonnet, and Gemini (±retrieval):
    - Claude gains 30+ points with retrieval, showing agent compatibility.
    - o3 excels on riddles; Gemini leads in CrossQA but lags on hard tasks.
- **Agents**: Evaluated ChatGPT Deep Research, Gemini Deep Research, and OpenHands Cloud on a hard subset, confirming ChatGPT's superiority but high latency.

#### 5. **Release** (Reviewer 2jv9)

- **Open-Source**: Will release immediately after NeurIPS announcement.
- **Test Set Improvements**: Acknowledged 275-sample limit (manual annotation cost) and will add confidence intervals and pass@k to measure variability.

### **Reviewer-Specific Responses & Improvements**

**Key Concerns Addressed**:

**5N3z**: SIS emergence validated via reward phase-out; confirmed WebPuzzle’s generation, curation, generalization.

**ucvZ**: Shared RL hyperparameters, SFT/RL data split; discussed DPO challenges and future work.

**2jv9**: Committed to open-sourcing; explained reward schedule; compared DeepDiver to ReAct; clarified search simulation limits.

**rFPC**: added closed-source model results; cross-benchmark comparisons; RL parameter details.

---

We believe these revisions enhance the clarity, rigor, and impact of our work. Thank you again for your invaluable guidance!

---

### Decision · Program_Chairs · 2025-09-17

**Decision:**

Accept (spotlight)

**Comment:**

This paper proposed a novel way to scale up the performance of search-based tasks in LLMs. It proposed a new dataset and evaluation metrics called WebPuzzle, which is carefully designed to test and induce model's effective use of search as a tool in RL training. The result shows good scaling of search behavior and competitive results.

Paper strength

Novel method that scales well in RL.
Thorough evaluation to show the scaling behavior.
Release of a new training and evaluation set WebPuzzle, which could be valuable to the broader community.

Paper weakness

Most of the weaknesses raised by the reviewers have been addressed in the rebuttal phase.
The authors acknowledged that the size of the evaluation dataset is on the smaller side, which is limited by cost of curation.

One Reviewer didn't respond to the rebuttal (rating=4). Upon close inspection, AC thinks all the concerns in the original review have been adequately addressed.

AC recommends acceptance of the paper based on the importance of the task, contribution of novel datasets, and nice thorough experiments to show the effectiveness.